# Quantifying fair income distribution in Thailand

**Thitithep Sitthiyot** [1] *, **Kanyarat Holasut** [2]

**1** Department of Banking and Finance, Faculty of Commerce and Accountancy, Chulalongkorn University, Bangkok, Thailand, **2** Department of Chemical Engineering, Faculty of Engineering, Khon Kaen University, Khon Kaen, Thailand

* thitithep@cbs.chula.ac.th

**Data Availability Statement:** All data used in this study are publicly available and can be accessed from the Office of the National Economic and Social Development Council website (https://www.nesdc.go.th/main.php?filename=PageSocial).

## Abstract

Given a vast concern about high income inequality in Thailand as opposed to empirical findings around the world showing people's preference for fair income inequality over unfair income equality, it is therefore important to examine whether inequality in income distribution in Thailand over the past three decades is fair, and what fair inequality in income distribution in Thailand should be. To quantitatively measure fair income distribution, this study employs the fairness benchmarks that are derived from the distributions of athletes' salaries in professional sports which satisfy the concepts of distributive justice and procedural justice, the no-envy principle of fair allocation, and the general consensus or the international norm criterion of a meaningful benchmark. By using the data on quintile income shares and the income Gini index of Thailand from the National Social and Economic Development Council, this study finds that, throughout the period from 1988 to 2021, the Thai income earners in the bottom 20%, the second 20%, and the top 20% receive income shares more than the fair shares whereas those in the third 20% and the fourth 20% receive income shares less than the fair shares. Provided that there are infinite combinations of quintile income shares that can have the same value of income Gini index but only one of them is regarded as fair, this study demonstrates the use of fairness benchmarks as a practical guideline for designing policies with an aim to achieve fair income distribution in Thailand. Moreover, a comparative analysis is conducted by employing the method for estimating optimal (fair) income distribution representing feasible income equality in order to provide an alternative recommendation on what optimal (fair) income distribution characterizing feasible income equality in Thailand should be.

## Introduction

Thailand has developed a society with high inequalities in income and wealth distributions [1]. Despite a remarkable progress in poverty reduction in the past three decades, income inequality in Thailand remains high [2]. With the income Gini index of 0.433 in 2019, Thailand has the highest income inequality in East Asia [2]. The income data published by the Office of the National Economic and Social Development Council (NESDC) [3] shows that, in 2021, the income earners in the highest 20% have income shares approximately 50% of total income

**Funding:** TS received funding for this study from Ratchadapiseksompotch Fund, Chulalongkorn University, Award Number: RCU_67_026_001. TS received salary from Chulalongkorn University (https://www.chula.ac.th/en/). The funder had no role in study design, data collection and analysis, decision to publish, or preparation of the manuscript. KS received no specific funding for this work.

**Competing interests:** The authors have declared that no competing interests exist.

share while those in the lowest 80% have income shares about 50% of total income share. This pattern of income distribution has not markedly changed since 1988 [4]. Credit Suisse [5] ranks Thailand as one of the most unequal countries in the world in terms of wealth inequality, with the top 1% holding 56% of total wealth. Laovakul [6] examines the concentration of titled land and other wealth in Thailand and finds that the wealth gap in Thailand is very highly concentrated as 80% of titled land is owned by the richest 5% and more than two-thirds of the country's assets is controlled by the richest 1%. Unequal patterns of ownership of land in turn lead to worsening economic and well-being conditions for poor farmers [7]. In addition, the United Nations Economic and Social Commission for Asia and Pacific [7] notes that income inequality causes a weakening of social bonds and an erosion of public trust in institutions which can raise social and political tensions and even lead to radicalization and crime. An empirical study by Lee et al. [8] shows that, in Asian countries including Thailand, an individual's perception of high income inequality lowers political trust but this linkage varies among countries with different degrees of income inequality. According to the United Nations [9], income inequality looms large and hinders Thailand's progress towards the Sustainable Development Goals (SDGs). As noted by the NESDC [10], the issue of income inequality in Thailand needs to be undertaken urgently in the Twelfth National Economic and Social Development Plan so that it does not become an obstacle to the country's economic and social development. One of policy targets as stated in the Twelfth National Economic and Social Development Plan is to reduce income inequality across the Thai population groups by lowering the income Gini index to 0.410 [10]. This policy target is in line with the Target 10.1 of the SDG 10 that calls for actions to lower inequality in income distribution within a country by gradually accomplishing and maintaining income growth of the lowest 40% at a rate that is higher than the country average by 2030 [11].

While there is a vast concern about high income inequality in Thailand, and reducing income inequality among different groups of population is considered an important goal as stated in Thailand Twelfth National Economic and Social Development Plan [10], a comprehensive study by Starmans et al. [12], drawing upon numerous studies in laboratory, cross-cultural research, as well as experiments with babies and young children, finds no evidence that people are bothered by income inequality itself but they are bothered by the issue that is often confounded with income inequality which is income unfairness. According to Starmans et al. [12], humans naturally favor fair income distributions, not equal ones, and when choosing between fair income inequality and unfair income equality, people prefer fair income inequality over unfair income equality. People even support substantial income inequality if they view as fair [13].

Given an immense concern about high income inequality in Thailand as opposed to empirical findings around the world showing people's preference for fair income inequality over unfair income equality, it is therefore important to examine whether inequality in income distribution in Thailand over the past three decades is fair, and what fair inequality in income distribution in Thailand should be. The findings from this study could enhance the understanding of income inequality and unfairness in income distribution which could benefit policymakers not only in evaluating the effectiveness of income redistributive measures but also in designing policies aimed to achieve fair income distribution across all population groups in Thailand.

While there exists a vast literature on fair inequality that provide a comprehensive theoretical and normative framework for measuring fair income distribution which allows for different viewpoints on what should be regarded as fair income distribution, for example, Almås et al. [14], Cappelen and Tungodden [15], and Hufe et al. [16], to our knowledge, there are a few studies that attempt to propose a benchmark for quantifying fair income distribution,

demonstrate the practical use of such a benchmark in order to gauge whether income distribution is fair, and provide a policy implication regarding how fair income distribution could be achieved based on the proposed benchmark. Those studies are Venkatasubramanian et al. [17], Park and Kim [18], and Sitthiyot and Holasut [19].

Venkatasubramanian et al. [17] propose a method for measuring how much inequality in income distribution is fair. Venkatasubramanian et al. [17]'s method is based on the microeconomic game theoretic framework and the concept of maximum entropy in statistical mechanics and information theory. Venkatasubramanian et al. [17] show that, in a perfectly competitive free market comprising self-interest utility-maximizing and profit-maximizing rational agents with negligible transaction costs, no illegal practices, no taxes, and no externalities, the fairest income distribution, when such a market is in equilibrium, is log-normal.

While Venkatasubramanian et al. [17]'s method has its own merit, this study would like to point out that the fairest income distribution being log-normal has a key limitation in that it characterizes an ideal free market operating under conditions that are not valid in real life as acknowledged by Venkatasubramanian et al. [17, p. 137]. Even though Venkatasubramanian et al. [17] find that, out of the sample of 12 developed countries, the empirical data on income distributions of four countries, namely, Norway, Sweden, Denmark, and Switzerland appear to be consistent with the outcome emerging from the ideal free market, the process of generating these outcomes in real life could differ since these countries operate under conditions that are totally different from what assumed in the microeconomic game theoretic model. Moreover, it is still debatable whether the distribution of income follows log-normal, gamma, power law [20], or a new type of distribution that has yet to be discovered. As noted by Chakrabarti et al. [21], scholars in economics, statistics, and physics tend to agree that the upper tail of income distribution could be well approximated by power law distribution but the lower tail of income distribution could be approximated by either one of these three distributions. For these reasons, all we can say is that log-normal distribution could be used to fit countries' income distributions. Whether the fairest distribution of income is log-normal in reality is not known since the exact process that generates this outcome is not specified in Venkatasubramanian et al. [17]'s study.

Park and Kim [18] develop a method for estimating feasible income equality that maximizes total social welfare. Recognizing that perfect income equality is idealistic and practically infeasible in reality, Park and Kim [18] demonstrate that optimal (fair) income distribution representing feasible income equality could be estimated by using the sigmoid function and the Boltzmann distribution. Park and Kim [18] reason that, in physical sciences, the Boltzmann distribution is based on entropy maximization and provides the most probable, natural, and unbiased distribution of a physical system which could be applied to analyze fair distribution of income in social sciences. Regarding the sigmoid function, Park and Kim [18] reason that it could reflect an increase in welfare as income rises in reality. According to Park and Kim [18], when income is below the critical low-income threshold, the level of welfare would slowly increase as income rises. This is because a rise in income is still not enough to support the basic needs. However, when income increases beyond the critical low-income threshold, the level of welfare would start to rise rapidly since the basic needs are fulfilled and, hence, there would be more economic freedom. As income increases further, the degree of economic freedom also increases but becomes saturated once it reaches the critical high-income threshold, and so does the level of welfare. Beyond the critical high-income threshold, the level of welfare would gradually increase as income increases. By modeling total social welfare based on the sigmoid function and income distribution based on the Boltzmann distribution, optimal (fair) income distribution representing feasible income equality could be estimated and compared to the actual income distribution in order to determine whether the existing income distribution is fair [18].

The empirical analysis conducted by Park and Kim [18] shows that the estimated optimal (fair) quintile income shares of four selected countries, namely, the United States of America (U.S.A.), China, Finland, and South Africa are very close to each other. So are the estimated values of corresponding income Gini index calculated based on the optimal (fair) income distributions of these four countries. Their empirical results lead Park and Kim [18] to conclude that it is possible to have a feasible income equality benchmark that is time-independent and universally applicable to all countries in the world. However, this study would like to note that, based on the estimated results on optimal (fair) quintile income shares and the corresponding income Gini index of China vs. those of Finland as reported in Park and Kim [18] which are 15.4% vs. 15.5% for the lowest quintile, 16.6% vs. 17.1% for the $2^{nd}$ quintile, 17.9% vs. 18.5% for the $3^{rd}$ quintile, 20.2% vs. 20.6% for the $4^{th}$ quintile, and 29.9% vs. 28.4% for the highest quintile, as well as 0.130 vs. 0.120 for the income Gini index, it is hard to imagine, both theoretically and empirically, that the optimal (fair) income shares by quintile and the corresponding income Gini index of China with population of 1.412 billion [22] would be very similar to those of Finland with population of 5.541 million [22]. According to Deltas [23], theoretically, the income Gini index of a country with small population would be smaller than that of a country with larger population generated by the same stochastic process. Deltas [23] also notes that, for any given level of intrinsic inequality as expressed by income generating function, a reduction in the sample size would lead to a reduction in inequality as measured by the income Gini index. Empirically, a study by Sitthiyot and Holasut [24] who examine the statistical relationship between the income Gini index and population size of 69 countries indicates that countries with small number of population tend to have small values of the income Gini index whereas larger values of the income Gini index are observed for countries with large number of population. For these reasons, this study views that the optimal (fair) income distribution representing feasible income equality that is constructed based on the sigmoid welfare function and the Boltzmann distribution proposed by Park and Kim [18] could be considered a proof of principle that fair income distribution can be quantified. It could be used as a practical guideline for government policy interventions if the policy target is to achieve a feasible income equality society, provided that perfect income equality is idealistic and practically infeasible in the real world [18].

Sitthiyot and Holasut [19] introduce a quantitative method for benchmarking fair income distribution for a particular value of income inequality as measured by the Gini index. Sitthiyot and Holasut [19]'s fairness benchmark is derived by using the actual data on distributions of athletes' salaries from various professional sports in order to estimate the statistical relationship between the quintile salary shares of professional athletes and the Gini index for professional athletes' salaries. According to Sitthiyot and Holasut [19], the rationale for using the distributions of actual athletes' salaries in professional sports is based on the ideas of distributive justice and procedural justice. For distributive justice, fairness is associated with the extent to which the outcomes of processes that allocate benefits and costs satisfy the equity rule which requires that individual and/or groups should receive benefits and costs in proportion to their contributions [25]. For procedural justice, fairness is associated with processes by which the authorities enact rules, resolve disputes, and allocate benefits and costs [25] and people have to agree that benefits and costs they receive result from fair processes with regard to benefits and costs allocations [13].

Provided that athletes' salaries in professional sports are allocated based on contributions of individual athletes and/or team efforts, i.e. distributive justice, who compete according to fair and transparent rules as well as codes of conduct that are written and administered by international sports authorities, i.e. procedural justice, all of which are understandable to and could be watched and judged by ordinary people all over the world as fair, regardless of their origins,

backgrounds, religious believes, or economic statuses, fairness benchmark that is derived based on the salaries of professional athletes therefore satisfy the no-envy principle of fair allocation [26] and the general consensus or the international norm criterion of a meaningful benchmark [27].

By using the data on income distribution of 75 countries, Sitthiyot and Holasut [19] demonstrate how the fairness benchmarks could be used to quantitatively measure whether quintile income shares of these countries are the fair shares for a particular value of the income Gini index, and what fair quintile income shares of these countries should be. The overall results indicate that the majority of income earners in the bottom 20%, the second 20% (35 countries), and the top 20% receive income shares more than the fair shares whereas income earners in the second 20% (40 countries), the third 20%, and the fourth 20% receive income shares less than the fair shares. Given that there are infinite combinations of quintile income shares that could have the same value of income Gini index but only one of them is regarded as fair, by using Sitthiyot and Holasut [19]'s fairness benchmarks as a guideline, different countries could choose different combinations of fair quintile income shares and the income Gini index that are suitable for their own contexts before designing policies in order to achieve fair income distributions across all population groups.

Based on the quantitative methods for benchmarking fair income distribution and their real-world applications as discussed above, along with the availability of data that can be used to conduct the analyses of fair income distribution in Thailand, this study chooses the method for benchmarking fair income distribution developed by Sitthiyot and Holasut [19] (SH method) in order to investigate whether inequality in income distribution in Thailand over the past three decades is fair, and to provide a practical guideline as to whether what fair income distribution among different groups of population in Thailand should be. In addition to SH method, this study conducts a comparative analysis of fair income distribution in Thailand by employing the method for estimating optimal (fair) income distribution representing feasible income equality proposed by Park and Kim [18] (PK method) in order to provide an alternative recommendation on what optimal (fair) income distribution characterizing feasible income equality in Thailand should be.

## Materials and methods

For notations, let $Q_i^S$, i = 1, 2, 3, 4, 5, be the quintile salary shares of professional athletes where $Q_1^S$ is salary share of the bottom 20%, $Q_2^S$ is salary share of the second 20%, $Q_3^S$ is salary share of the third 20%, $Q_4^S$ is salary share of the fourth 20%, and $Q_5^S$ is salary share of the top 20%, respectively. Also, let $Gini_S$ be the Gini index for professional athletes' salaries. Sitthiyot and Holasut [19] derive the fairness benchmark by employing the data on annual salaries of athletes from 11 well-known professional sports from Sitthiyot [28]. Those 11 professional sports are the Women's National Basketball Association, the English Premier League, the National Football League, the National Hockey League, the Major League Baseball, the National Basketball Association, the Professional Golfers' Association of America, the Ladies Professional Golf Association, the Major League Soccer, the Association of Tennis Professionals, and the Women's Tennis Association. According to Sitthiyot and Holasut [19], these data do not come from one specific group of population and/or one particular country, and therefore could avoid or mitigate the problem of selection bias since the athletes competing in these professional sports come from diverse ethnic, social, national, and regional backgrounds. In addition, these professional sports tournaments are organized in different countries around the world. Furthermore, Sitthiyot and Holasut [19] note that rules, regulations, and player's codes of conduct in

these professional sports are written and administered by relevant international sports authorities whose members are from different countries.

By using the actual data on athletes' salaries from 11 professional sports, Sitthiyot and Holasut [19] first construct the Lorenz curve for each professional sport. The constructed Lorenz curve is then used for calculating $Q_i^S$s and $Gini_S$ for each professional sport. Thus, there is an ordered pair of $Gini_S$ and $Q_i^S$ for each professional sport, totaling 11 ordered pairs for each quintile. To derive fairness benchmarks, one for each quintile, Sitthiyot and Holasut [19] estimate the statistical relationships between $Q_i^S$s and $Gini_S$s, where $0 < Gini_S < 1$. That is, for a particular value of $Gini_S$, where $0 < Gini_S < 1$, there would be only one combination of $Q_i^S$s that is regarded as fair. In order to guarantee that fairness benchmarks satisfy the mathematical properties of the Lorenz curve and the Gini index, Sitthiyot and Holasut [19] impose five conditions when estimating the relationships between $Q_i^S$s and $Gini_S$s. The first condition is that the fairness benchmark for the 5th quintile must pass two coordinates which are (0, 0.2) and (1, 1). The rationale for the first condition is that when $Gini_S$ is equal to 0, $Q_5^S$ must be equal to 0.2, and when $Gini_S$ is equal to 1, $Q_5^S$ must be equal to 1. For the second condition, the fairness benchmarks for the other four quintiles must pass two coordinates which are (0, 0.2) and (1, 0). The justification for the second condition is that when $Gini_S$ equals 0, $Q_1^S$, $Q_2^S$, $Q_3^S$ and $Q_4^S$ all have to be equal to 0.2, and when $Gini_S$ is equal to 1, $Q_1^S$, $Q_2^S$, $Q_3^S$ and $Q_4^S$ all have to be equal to 0. Given that the Lorenz curve must be a monotonically increasing function, the third condition is that, for a particular value of $Gini_S$, $0 \leq Q_1^S \leq Q_2^S \leq Q_3^S \leq Q_4^S \leq Q_5^S \leq 1$. For the fourth condition, $\frac{dQ_1^s}{dGini_s}\Big|_{Gini_S=0} \leq \frac{dQ_2^s}{dGini_s}\Big|_{Gini_S=0} \leq \frac{dQ_3^s}{dGini_s}\Big|_{Gini_S=0} \leq \frac{dQ_4^s}{dGini_s}\Big|_{Gini_S=0} \leq \frac{dQ_5^s}{dGini_s}\Big|_{Gini_S=0}$. That is when $Gini_S$ is equal to 0, the slope of fairness benchmark for the 1st quintile $\leq$ the slope of fairness benchmark for the 2nd quintile $\leq$ the slope of fairness benchmark for the 3rd quintile $\leq$ the slope of fairness benchmark for the 4th quintile $\leq$ the slope of fairness benchmark for the 5th quintile. With regard to the fifth condition, $\sum_{i=1}^{5} Q_i^S$ for each professional sport must equal 1, for a particular value of $Gini_S$.

Given these five conditions, the fairness benchmarks representing the statistical relationships between $Q_i^S$s and $Gini_S$s, where $0 < Gini_S < 1$, which satisfy the notions of procedural justice and distributive justice, the no-envy principle of fair allocation [26], and the general consensus or the international norm criterion of a meaningful benchmark [27] in that the distributions of professional athletes' salaries are, by and large, the products of fair rules, individual, and/or team performance, are shown as Eqs (1)–(5).

$$Q_1^S = 0.1956Gini_S^4 - 0.6612Gini_S^3 + 0.9356Gini_S^2 - 0.6700Gini_S + 0.20 \qquad (1)$$

$$Q_2^S = -0.4914Gini_S^4 + 1.3968Gini_S^3 - 1.1195Gini_S^2 + 0.0141Gini_S + 0.20 \qquad (2)$$

$$Q_3^S = 0.2545Gini_S^4 + 0.0508Gini_S^3 - 0.6650Gini_S^2 + 0.1597Gini_S + 0.20 \qquad (3)$$

$$Q_4^S = 1.9481Gini_S^4 - 3.3533Gini_S^3 + 1.0625Gini_S^2 + 0.1428Gini_S + 0.20 \qquad (4)$$

$$Q_5^S = -1.9067Gini_S^4 + 2.5669Gini_S^3 - 0.2136Gini_S^2 + 0.3534Gini_S + 0.20 \qquad (5)$$

Note that, in order to use Eqs (1)–(5) for measuring fair income distribution, this study assumes that the distributions of professional athletes' salaries are stable across time. Fig 1 illustrates the fairness benchmarks representing the statistical relationships between $Q_i^S$s and $Gini_S$s, where $0 < Gini_S < 1$, for each quintile.

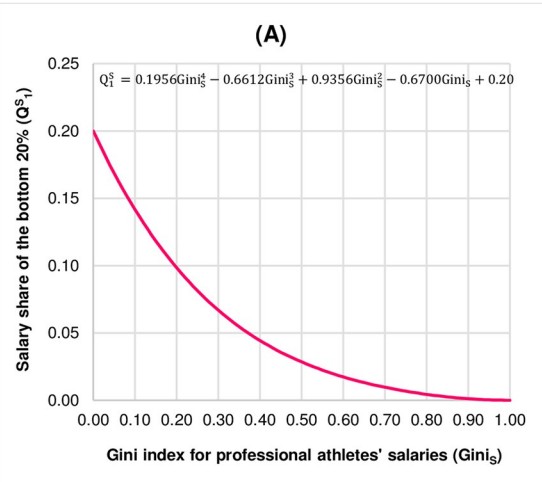

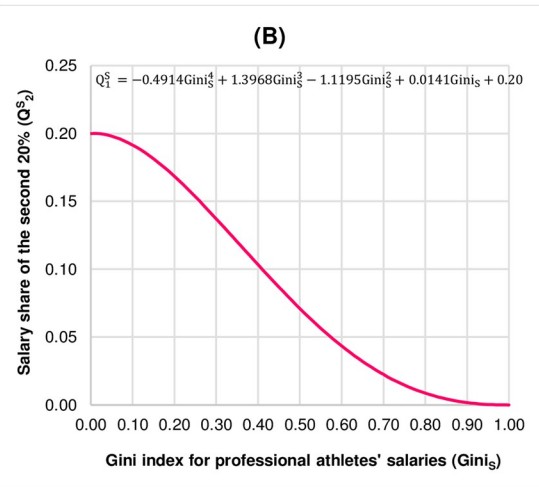

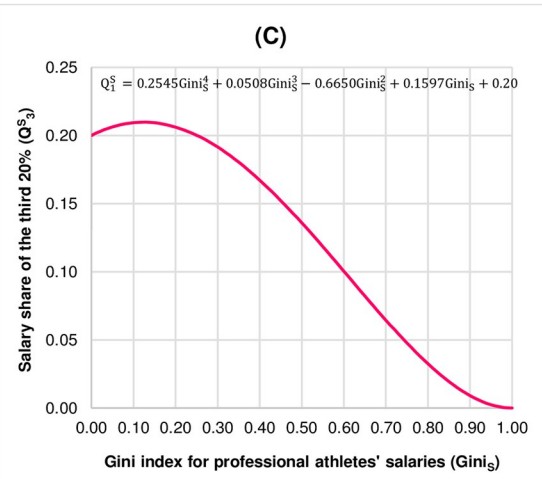

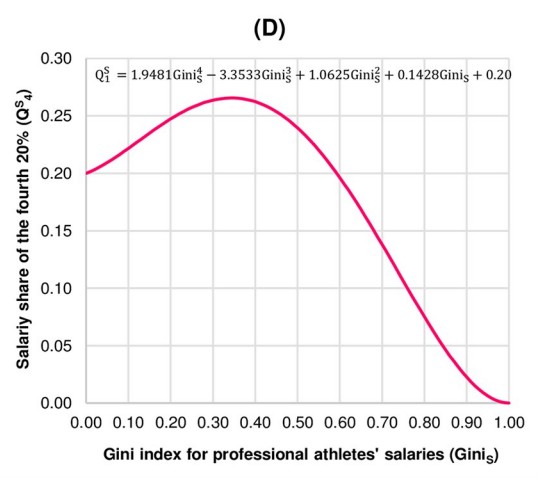

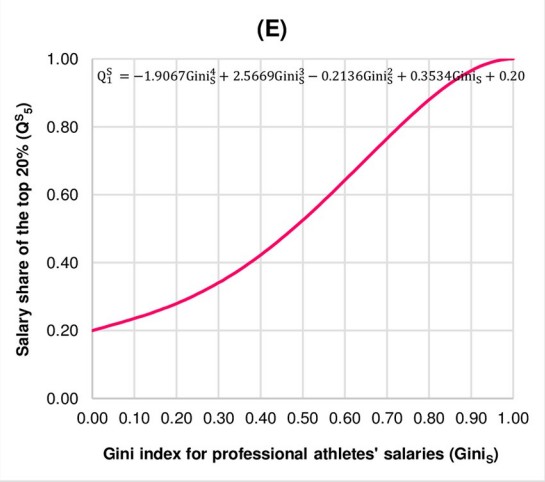

**Fig 1. Fairness benchmarks representing the statistical relationships between the quintile salary shares of professional athletes** $(Q_i^S, i = 1, 2, 3, 4, 5)$ **and the Gini index for professional athletes' salaries (Gini$_S$), where $0 <$ Gini$_S < 1$.** (A) Bottom 20%. (B) Second 20%. (C) Third 20%. (D) Fourth 20%. (E) Top 20%. Note that the scale of $Q_1^S$, $Q_2^S$, and $Q_3^S$ is between 0.00 and 0.25. The scale of $Q_4^S$ is between 0.00 and 0.30 while the scale of $Q_5^S$ is between 0.00 and 1.00.

According to Sitthiyot and Holasut [19], the way in which each fairness benchmark is used to measure whether income distribution by quintile is fair is by plotting the Cartesian coordinate where the abscissa is the income Gini index and the ordinate is the quintile income share, denoted as $Q_i^I$, i = 1, 2, 3, 4, 5, and then comparing it to its corresponding fairness benchmark in order to determine whether the ordered pair (income Gini index, $Q_i^I$) is on, above, or below the corresponding fairness benchmark for a particular value of income Gini index. The closer the ordered pair (income Gini index, $Q_i^I$) is to the fairness benchmark, the fairer the income share in that quintile. If the ordered pair (income Gini index, $Q_i^I$) is above the fairness benchmark, it suggests that the income earners in that quintile receive income share more than the fair share. In contrast, if the ordered pair (income Gini index, $Q_i^I$) is below the fairness benchmark, it indicates that the income earners in that quintile receive income share less than the fair share.

In this study, the degree of unfairness in quintile income shares can be quantified by computing the difference between actual quintile income share and fair quintile income share based on SH method, denoted as Δ, for a particular value of income Gini index. Relative to the fair quintile income share, the closer the value of Δ is to zero, the fairer the quintile income share whereas the farther the value of Δ is from zero in either positive or negative direction, the more unfair the quintile income share. Provided that there are infinite combinations of quintile income shares that can have an identical value of income Gini index but there is only one of them that is regarded as fair, the fairness benchmarks based on SH method can thus be used to provide a practical guideline on what fair quintile income shares should be for a particular value of income Gini index.

In addition to SH method, this study conducts a comparative analysis by employing PK method in order to measure whether income distribution in Thailand is fair. Based on Park and Kim [18], optimal (fair) income distributions representing feasible income equality could be modeled by the sigmoid function and the Boltzmann distribution as discussed in Introduction. For notations, let $Q_i^I$ be actual quintile income share of population group i, $y_i$ be quintile income share of population group i based on the Boltzmann distribution, and β be a parameter, where $0 \leq \beta \leq 1$. According to Park and Kim [18], in the Boltzmann distribution, the probability ($P_i$) that a unit income is distributed to quintile population group i can be shown as Eq 6.

$$P_i = \frac{e^{\beta Q_i^I}}{\sum_{i=1}^{5} e^{\beta Q_i^I}}, \; i = 1, \, 2, \, 3, \, 4, \, 5 \tag{6}$$

Let Y be total income. Park and Kim [18] set the value of Y to be 100. The quintile income share distributed to quintile population group i based on the Boltzmann distribution ($y_i$) can be computed as shown in Eq 7.

$$y_i = Y * \frac{e^{\beta Q_i^I}}{\sum_{i=1}^{5} e^{\beta Q_i^I}}, i = 1, \, 2, \, 3, \, 4, \, 5 \text{ and } Y = 100 \tag{7}$$

As stated by Park and Kim [18], when $y_i$s are inserted into the sigmoid total social welfare function (W), it becomes a function of β as shown in Eq 8.

$$\text{Max}_\beta W(y_1, y_2, y_3, y_4, y_5) = \sum_{i=1}^{5} \frac{1}{\left(1 + e^{\alpha * (\mu - y_i)}\right)}, \tag{8}$$

given

$$y_i = Y * \frac{e^{\beta Q_i^I}}{\sum_{i=1}^{5} e^{\beta Q_i^I}}, \; i = 1, \, 2, \, 3, \, 4, \, 5$$

Note that the parameters μ and α are the critical low-income share threshold and the critical high-income share threshold which are defined as $\frac{(L+H)}{2}$ and $\frac{6}{(H-L)}$, respectively, where $L = \frac{Q_2^I + Q_3^I}{2}$ and $H = \frac{Q_4^I + Q_5^I}{2}$ [18]. Provided that W can be maximized at a specific value of β denoted as β*, the corresponding values of $Q_i^I$s with β* would represent the optimal (fair) income distribution.

Similar to SH method, the way in which PK method is used to determine whether income distribution is fair is by comparing the actual quintile income share to its respective optimal (fair) quintile income share. The closer the actual quintile income share to the optimal (fair) quintile income share, the fairer the income share in that quintile. If the actual quintile income share is higher than the optimal (fair) quintile income share, it indicates that the income earners in that quintile receive income share more than the optimal (fair) share. In contrast, if the actual quintile income share is lower than the optimal (fair) quintile income share, it suggests that the income earners in that quintile receive income share less than the optimal (fair) share. Park and Kim [18] measures the degree of unfairness in quintile income share by calculating the difference between the actual quintile income share and the optimal (fair) quintile income share, also denoted as Δ. The closer the value of Δ is to 0, the fairer the income share in that quintile whereas the farther the value of Δ is from 0 in either positive or negative direction, the more unfair the income share in that quintile.

To analyze whether inequality in income distribution in Thailand over the past three decades is fair, this study employs the data on quintile income shares and income Gini index of Thailand from 1988 to 2021 for plotting the ordered pair (income Gini index, $Q_i^I$) which would then be used to compare to its respective fairness benchmark based on SH method. The data on income shares by quintile of Thailand from the same period are also used to compare to their corresponding optimal (fair) quintile income shares estimated according to PK method. All data are publicly available and can be accessed from the NESDC [3]. Provided that the data on quintile income shares and income Gini index are reported once every two years, there are total of 18 years of observations. Table 1 reports the descriptive statistics of data on quintile income shares and income Gini index of Thailand from 1988 to 2021.

## Results

### Fair income distribution based on SH method

The results on scatter plots of the Cartesian coordinates of income Gini index and quintile income shares ($Q_i^I$, i = 1, 2, 3, 4, 5) of Thailand from 1988 to 2021 and their corresponding fairness benchmarks for each quintile are illustrated in Fig 2.

**Table 1. The descriptive statistics of data on quintile income shares and income Gini index of Thailand from 1988 to 2021.**

| Quintile income shares and income Gini index | Mean | Median | Mode | Minimum | Maximum | Standard deviation | No. of observations |
|---|---|---|---|---|---|---|---|
| Income share held by the bottom 20% | 0.045 | 0.043 | - | 0.038 | 0.055 | 0.005 | 18 |
| Income share held by the second 20% | 0.082 | 0.080 | - | 0.071 | 0.097 | 0.008 | 18 |
| Income share held by the third 20% | 0.126 | 0.124 | - | 0.111 | 0.143 | 0.010 | 18 |
| Income share held by the fourth 20% | 0.203 | 0.202 | - | 0.189 | 0.217 | 0.007 | 18 |
| Income share held by the top 20% | 0.545 | 0.548 | - | 0.489 | 0.590 | 0.029 | 18 |
| Income Gini index | 0.489 | 0.496 | - | 0.429 | 0.536 | 0.032 | 18 |

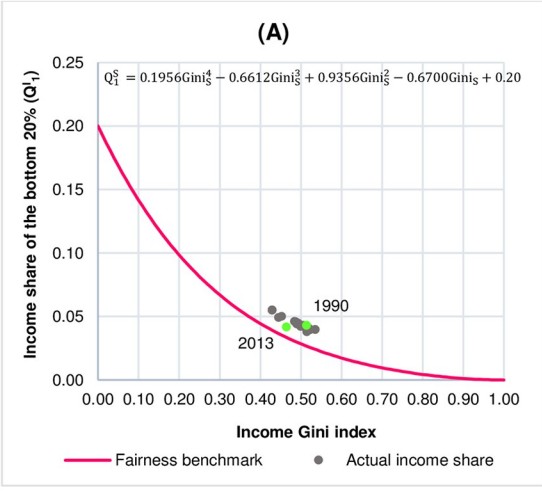

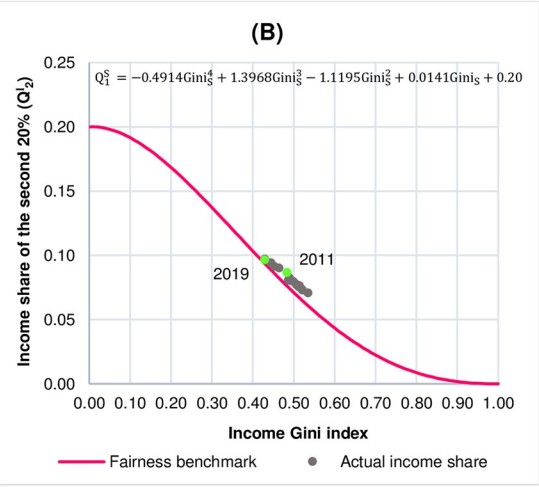

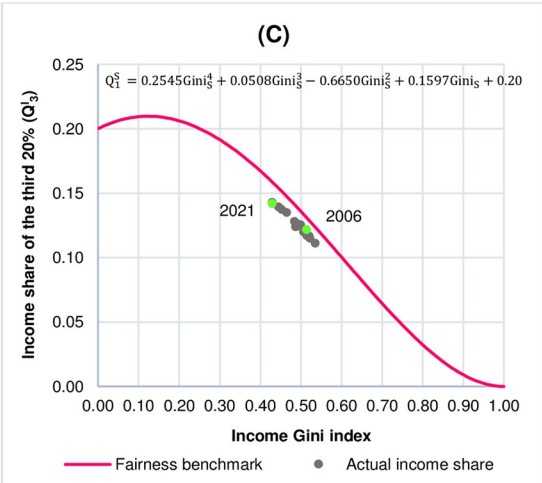

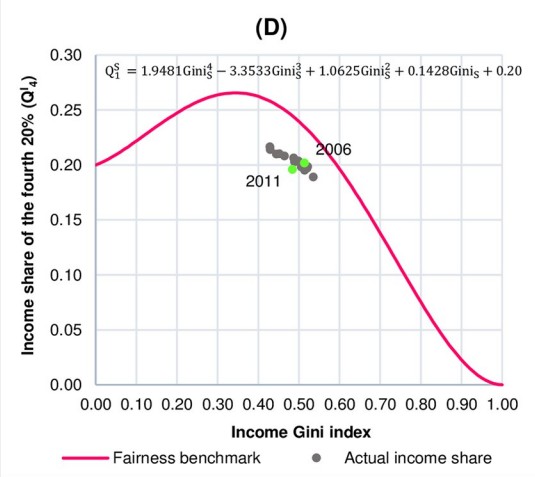

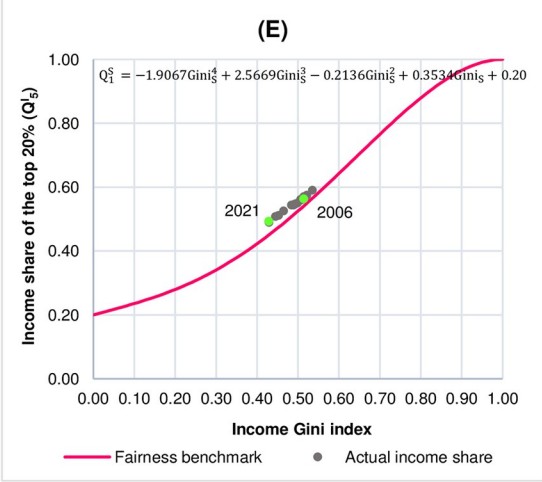

**Fig 2. Scatter plots of the Cartesian coordinates of income Gini index and quintile income shares** ($Q_i^I$, **i = 1, 2, 3, 4, 5) of Thailand from 1988 to 2021.** (A) Bottom 20% ($Q_1^I$). (B) Second 20% ($Q_2^I$). (C) Third 20% ($Q_3^I$). (D) Fourth 20% ($Q_4^I$). (E) Top 20% ($Q_5^I$). Note that the scale of $Q_1^I$, $Q_2^I$, and $Q_3^I$ is between 0.00 and 0.25. The scale of $Q_4^I$ is between 0.00 and 0.30 while the scale of $Q_5^I$ is between 0.00 and 1.00.

The overall results indicate that the ordered pairs (income Gini index, $Q_i^I$) are either above or below the fairness benchmarks in all five quintiles, with those in the bottom, the 2nd, and the top quintile being above their corresponding fairness benchmarks and those in the 3rd and the 4th quintile being below their corresponding fairness benchmarks. These results indicate that, relative to the fairness benchmarks, the Thai income earners in the bottom, the 2nd, and the top quintile receive income shares higher than the fair shares whereas the Thai income earners in the 3rd and the 4th quintile receive income shares lower than the fair shares.

Given that the degree of unfairness in income shares by quintile is measured by the difference between actual quintile income shares and fair quintile income shares based on SH method ($\Delta$), our results, as shown in Fig 2 and also reported in Table 2, indicate that, for the Thai income earners in the bottom, the 2nd, and the top quintile, all of which receive income shares higher than the fair shares, the income earners in the top quintile receive income share higher than the fair share the most ($\Delta = 0.041$) whereas, for the Thai income earners in the 3rd and the 4th quintile, both of which receive income shares lower than the fair shares, the income earners in the 4th quintile receive income share lower than the fair share the most ($\Delta = -0.049$).

Considering the degree of unfair income shares in each quintile as shown in Fig 2 and in Table 2, the results from the 1st quintile, where the income share is held by the bottom 20% (Fig 2A), suggest that the income share the income earners receive in 1990 is higher than the fair share the most ($\Delta = 0.016$) whereas the income share the income earners receive in 2013 is higher than the fair share the least ($\Delta = 0.008$). For the 2nd quintile where income share is held by the second 20% (Fig 2B), the results indicate that, in 2011, the income earners receive income share more than the fair share the most ($\Delta = 0.011$) while, in 2019, the income earners receive income share more than the fair share the least ($\Delta = 0.003$). The results from the 3rd quintile, where income share is held by the third 20% (Fig 2C), show that, in 2021, the income earners receive income share lower than the fair share the most ($\Delta = -0.017$) whereas, in 2006, the income earners receive income share lower than the fair share the least ($\Delta = -0.009$). For the 4th quintile where the income share is held by the fourth 20% (Fig 2D), the results show that, in 2011, the income earners receive income share less than the fair share the most ($\Delta = -0.049$) while, in 2006, the income earners receive income share less than the fair share the least ($\Delta = -0.033$). Lastly, the results for the 5th quintile, where the income share is held by the top 20% (Fig 2E), indicate that the year that the income earners receive income share more than the fair share the most is 2021 ($\Delta = 0.041$) while the year that the income earners receive income share more than the fair share the least is 2006 ($\Delta = 0.021$). The actual quintile income shares vs. the fair quintile income shares based on SH method as well as the values of income Gini index of Thailand from 1988 to 2021 are reported in Table 2. Fig 3 illustrates the degree of unfairness in quintile income shares of Thailand from 1988 to 2021 as measured by the difference between actual quintile income shares and fair quintile income shares ($\Delta$) according to SH method. As explained in Materials and Methods, relative to its respective fairness benchmark, the closer the value of $\Delta$ is to zero, the fairer the quintile income share whereas the farther the value of $\Delta$ is from zero in either positive or negative direction, the more unfair the quintile income share.

As shown in Fig 3, the degrees of unfairness in quintile income shares of the Thai income earners in the top 20% and the fourth 20% are rising as shown by the values of $\Delta$s that are further way from 0 whereas those in the bottom 20% and the third 20% remains mostly unchanged throughout the period. For the Thai income earners in the second 20%, the degree of unfairness in quintile income share decreases slightly as shown by the decreasing value of $\Delta$.

**Table 2. Thailand's actual quintile income shares, fair quintile income shares based on SH method, optimal (fair) quintile income shares representing feasible income equality based on PK method, and their corresponding values of income Gini index.**

| Year | Income share | Bottom 20% | Δ | Second 20% | Δ | Third 20% | Δ | Fourth 20% | Δ | Top 20% | Δ | Income Gini index |
|------|-------------|-----------|-----|-----------|-----|----------|-----|-----------|-----|--------|-----|------------------|
| 1988 | Actual | 0.046 | | 0.080 | | 0.124 | | 0.206 | | 0.544 | | 0.487 |
| | SH method | 0.030 | 0.016 | 0.075 | 0.006 | 0.140 | -0.016 | 0.244 | -0.037 | 0.511 | 0.033 | 0.487 |
| | PK method | 0.156 | -0.111 | 0.164 | -0.084 | 0.174 | -0.050 | 0.195 | 0.011 | 0.310 | 0.234 | 0.135 |
| 1990 | Actual | 0.043 | | 0.075 | | 0.117 | | 0.195 | | 0.570 | | 0.515 |
| | SH method | 0.027 | 0.016 | 0.067 | 0.009 | 0.131 | -0.014 | 0.235 | -0.039 | 0.541 | 0.028 | 0.515 |
| | PK method | 0.158 | -0.115 | 0.165 | -0.090 | 0.174 | -0.057 | 0.192 | 0.003 | 0.310 | 0.259 | 0.133 |
| 1992 | Actual | 0.040 | | 0.071 | | 0.111 | | 0.189 | | 0.590 | | 0.536 |
| | SH method | 0.024 | 0.016 | 0.061 | 0.010 | 0.124 | -0.012 | 0.226 | -0.037 | 0.566 | 0.024 | 0.536 |
| | PK method | 0.159 | -0.119 | 0.165 | -0.094 | 0.173 | -0.062 | 0.191 | -0.002 | 0.312 | 0.278 | 0.133 |
| 1994 | Actual | 0.041 | | 0.074 | | 0.117 | | 0.197 | | 0.572 | | 0.520 |
| | SH method | 0.026 | 0.015 | 0.065 | 0.009 | 0.129 | -0.012 | 0.232 | -0.035 | 0.548 | 0.024 | 0.520 |
| | PK method | 0.157 | -0.117 | 0.164 | -0.091 | 0.174 | -0.057 | 0.193 | 0.004 | 0.312 | 0.260 | 0.135 |
| 1996 | Actual | 0.042 | | 0.076 | | 0.118 | | 0.199 | | 0.565 | | 0.513 |
| | SH method | 0.027 | 0.015 | 0.067 | 0.008 | 0.132 | -0.013 | 0.235 | -0.036 | 0.539 | 0.026 | 0.513 |
| | PK method | 0.157 | -0.115 | 0.164 | -0.089 | 0.174 | -0.055 | 0.193 | 0.006 | 0.312 | 0.254 | 0.135 |
| 1998 | Actual | 0.043 | | 0.078 | | 0.120 | | 0.198 | | 0.561 | | 0.507 |
| | SH method | 0.028 | 0.015 | 0.069 | 0.008 | 0.134 | -0.014 | 0.237 | -0.039 | 0.532 | 0.029 | 0.507 |
| | PK method | 0.157 | -0.114 | 0.165 | -0.087 | 0.174 | -0.054 | 0.193 | 0.005 | 0.311 | 0.251 | 0.134 |
| 2000 | Actual | 0.039 | | 0.073 | | 0.115 | | 0.198 | | 0.574 | | 0.522 |
| | SH method | 0.026 | 0.014 | 0.065 | 0.008 | 0.128 | -0.013 | 0.232 | -0.033 | 0.550 | 0.024 | 0.522 |
| | PK method | 0.157 | -0.118 | 0.164 | -0.091 | 0.173 | -0.058 | 0.193 | 0.006 | 0.313 | 0.261 | 0.136 |
| 2002 | Actual | 0.042 | | 0.077 | | 0.121 | | 0.201 | | 0.560 | | 0.508 |
| | SH method | 0.027 | 0.014 | 0.069 | 0.008 | 0.133 | -0.013 | 0.237 | -0.036 | 0.534 | 0.026 | 0.508 |
| | PK method | 0.157 | -0.115 | 0.164 | -0.087 | 0.174 | -0.053 | 0.194 | 0.008 | 0.312 | 0.248 | 0.136 |
| 2004 | Actual | 0.045 | | 0.080 | | 0.125 | | 0.203 | | 0.547 | | 0.493 |
| | SH method | 0.029 | 0.015 | 0.073 | 0.007 | 0.138 | -0.014 | 0.242 | -0.039 | 0.517 | 0.030 | 0.493 |
| | PK method | 0.157 | -0.112 | 0.164 | -0.084 | 0.175 | -0.050 | 0.194 | 0.009 | 0.310 | 0.237 | 0.135 |
| 2006 | Actual | 0.038 | | 0.076 | | 0.122 | | 0.202 | | 0.562 | | 0.514 |
| | SH method | 0.027 | 0.011 | 0.067 | 0.010 | 0.131 | -0.009 | 0.235 | -0.033 | 0.541 | 0.021 | 0.514 |
| | PK method | 0.155 | -0.117 | 0.163 | -0.087 | 0.174 | -0.052 | 0.193 | 0.008 | 0.315 | 0.248 | 0.140 |
| 2007 | Actual | 0.042 | | 0.080 | | 0.125 | | 0.204 | | 0.549 | | 0.499 |
| | SH method | 0.029 | 0.013 | 0.071 | 0.008 | 0.136 | -0.011 | 0.240 | -0.036 | 0.524 | 0.025 | 0.499 |
| | PK method | 0.155 | -0.113 | 0.164 | -0.084 | 0.174 | -0.049 | 0.194 | 0.009 | 0.312 | 0.237 | 0.138 |
| 2009 | Actual | 0.044 | | 0.083 | | 0.127 | | 0.203 | | 0.544 | | 0.490 |
| | SH method | 0.030 | 0.014 | 0.074 | 0.008 | 0.139 | -0.013 | 0.243 | -0.040 | 0.514 | 0.030 | 0.490 |
| | PK method | 0.156 | -0.112 | 0.164 | -0.082 | 0.175 | -0.048 | 0.194 | 0.009 | 0.311 | 0.233 | 0.136 |
| 2011 | Actual | 0.046 | | 0.086 | | 0.128 | | 0.196 | | 0.544 | | 0.484 |
| | SH method | 0.031 | 0.015 | 0.076 | 0.011 | 0.141 | -0.013 | 0.245 | -0.049 | 0.508 | 0.036 | 0.484 |
| | PK method | 0.157 | -0.111 | 0.166 | -0.080 | 0.176 | -0.048 | 0.193 | 0.003 | 0.309 | 0.235 | 0.132 |
| 2013 | Actual | 0.042 | | 0.090 | | 0.135 | | 0.208 | | 0.525 | | 0.465 |
| | SH method | 0.033 | 0.008 | 0.082 | 0.008 | 0.148 | -0.012 | 0.250 | -0.042 | 0.487 | 0.038 | 0.465 |
| | PK method | 0.152 | -0.111 | 0.164 | -0.074 | 0.175 | -0.040 | 0.195 | 0.013 | 0.313 | 0.212 | 0.142 |
| 2015 | Actual | 0.049 | | 0.094 | | 0.139 | | 0.210 | | 0.507 | | 0.445 |
| | SH method | 0.037 | 0.013 | 0.088 | 0.006 | 0.154 | -0.014 | 0.255 | -0.045 | 0.466 | 0.041 | 0.445 |
| | PK method | 0.154 | -0.105 | 0.165 | -0.071 | 0.176 | -0.037 | 0.196 | 0.014 | 0.309 | 0.199 | 0.137 |

*(Continued)*

**Table 2.** (Continued)

| Year | Income share | Bottom 20% | Δ | Second 20% | Δ | Third 20% | Δ | Fourth 20% | Δ | Top 20% | Δ | Income Gini index |
|------|-------------|-----------|------|-----------|-------|----------|--------|-----------|--------|---------|-------|-------------------|
| 2017 | Actual | 0.050 | | 0.091 | | 0.137 | | 0.210 | | 0.511 | | 0.453 |
| | SH method | 0.035 | 0.015 | 0.086 | 0.006 | 0.151 | -0.014 | 0.253 | -0.043 | 0.474 | 0.037 | 0.453 |
| | PK method | 0.155 | -0.104 | 0.165 | -0.073 | 0.176 | -0.039 | 0.196 | 0.014 | 0.308 | 0.203 | 0.136 |
| 2019 | Actual | 0.055 | | 0.096 | | 0.143 | | 0.217 | | 0.489 | | 0.429 |
| | SH method | 0.039 | 0.016 | 0.094 | 0.003 | 0.159 | -0.016 | 0.258 | -0.041 | 0.451 | 0.039 | 0.429 |
| | PK method | 0.154 | -0.099 | 0.164 | -0.068 | 0.177 | -0.034 | 0.199 | 0.018 | 0.306 | 0.183 | 0.136 |
| 2021 | Actual | 0.055 | | 0.097 | | 0.142 | | 0.214 | | 0.492 | | 0.430 |
| | SH method | 0.039 | 0.016 | 0.093 | 0.004 | 0.159 | -0.017 | 0.258 | -0.044 | 0.451 | 0.041 | 0.430 |
| | PK method | 0.154 | -0.099 | 0.165 | -0.068 | 0.177 | -0.035 | 0.198 | 0.016 | 0.306 | 0.187 | 0.134 |

The income Gini index and the quintile income shares are in decimals.

## Optimal (fair) income distribution based on PK method

Next, this study reports the results on comparative analysis of fair income distribution in Thailand based on PK method. Fig 4 illustrates the actual quintile income shares of Thailand from 1988 to 2021 and their corresponding optimal (fair) quintile income shares representing feasible income equality. Note that the results on the calculated values of L, H, μ and α as well as the estimated values of W and β* are reported in S1 Table.

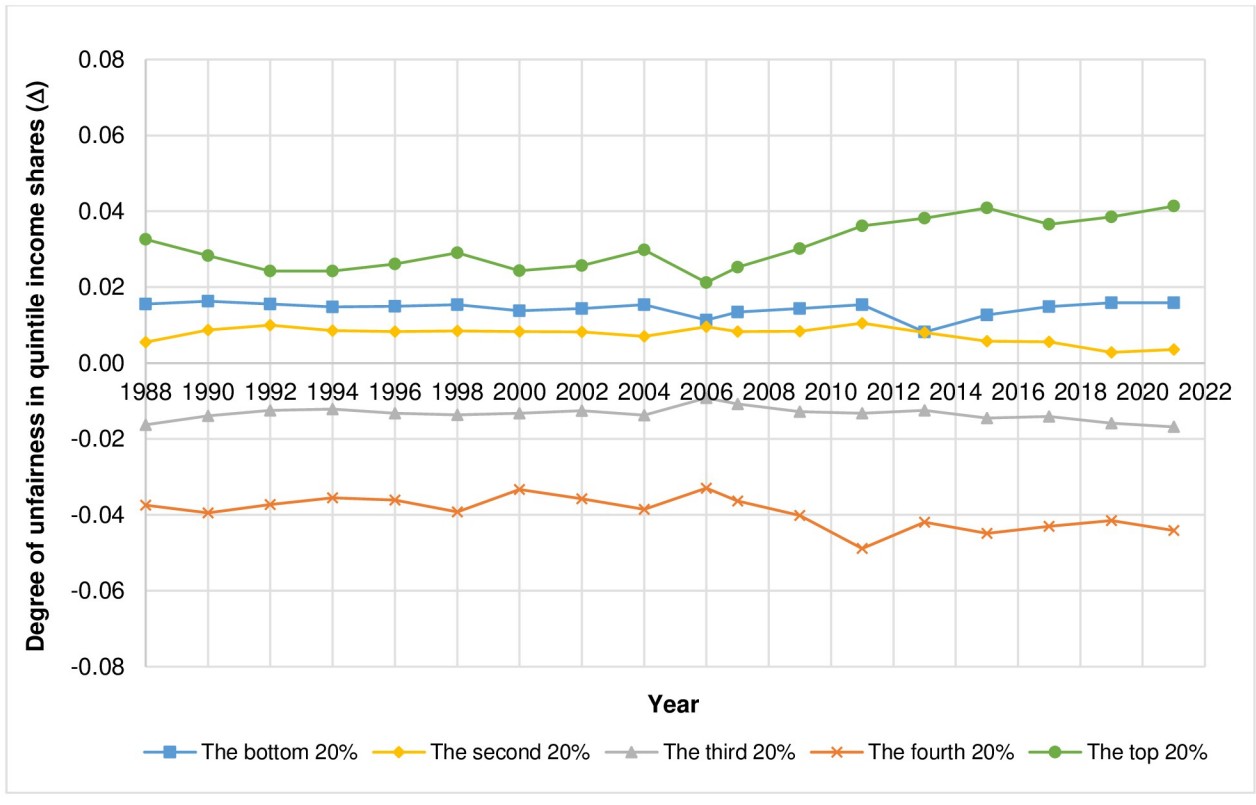

**Fig 3. The degree of unfairness in quintile income shares of Thailand from 1988 to 2021 as measured by the difference between actual quintile income shares and fair quintile income shares (Δ) according to SH method.** Note that the scale of Δ is between– 0.08 and 0.08.

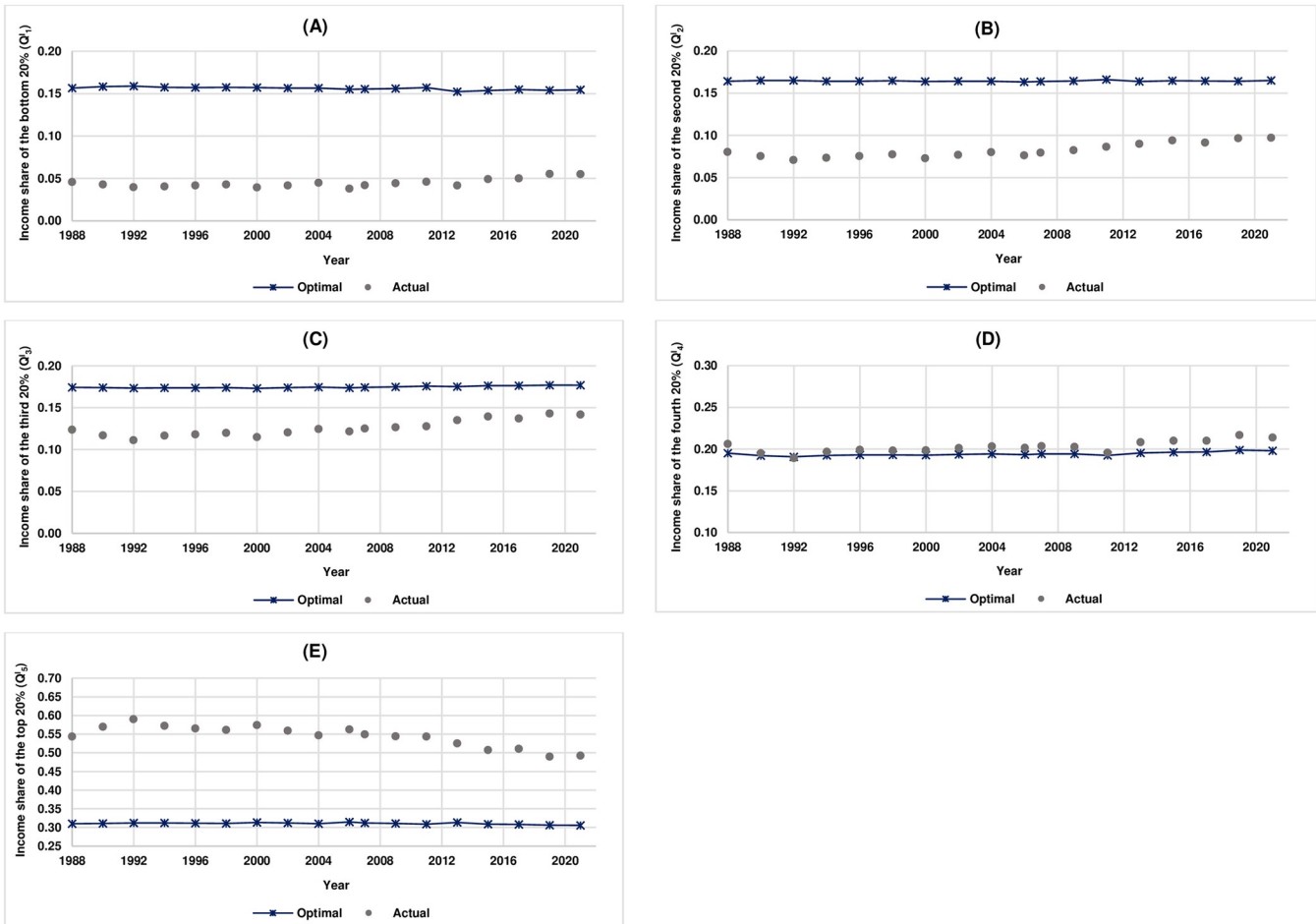

**Fig 4. Actual quintile income shares of Thailand and their corresponding optimal (fair) quintile income shares from 1988 to 2021.** (A) Bottom 20%. (B) Second 20%. (C) Third 20%. (D) Fourth 20%. (E) Top 20%. Note that the scale of $Q_1^I$, $Q_2^I$, and $Q_3^I$ is between 0.00 and 0.20. The scale of $Q_4^I$ is between 0.10 and 0.30 while the scale of $Q_5^I$ is between 0.25 and 0.70.

The results, as shown in Fig 4 and also reported in Table 2, indicate that the actual income shares of the bottom, the 2nd, and the 3rd quintile are lower than the optimal (fair) income shares while those of the 4th and the top quintile are higher than the optimal (fair) income shares. These results suggest that, according to PK method, the Thai income earners in the bottom, the 2nd, and the 3rd quintile receive income shares less than the optimal (fair) shares, with those in the bottom quintile receive income share less than the optimal (fair) share the most (Δ = −0.119), whereas the Thai income earners in the 4th and the top quintile receive income shares more than the optimal (fair) shares, with those in the top quintile receive income share more than the optimal (fair) share the most (Δ = 0.279). Fig 5 illustrates the degree of unfairness in quintile income shares of Thailand from 1988 to 2021 as measured by the difference between actual quintile income shares and optimal (fair) quintile income shares (Δ) based on PK method. The closer the value of Δ is to zero, the fairer the quintile income share whereas the further the value of Δ is away from zero in either positive or negative direction, the more unfair the quintile income share.

As illustrated in Fig 5, the degrees of unfairness in quintile income shares of the Thai income earners in the bottom 20%, the second 20%, the third 20% and the top 20% are slightly

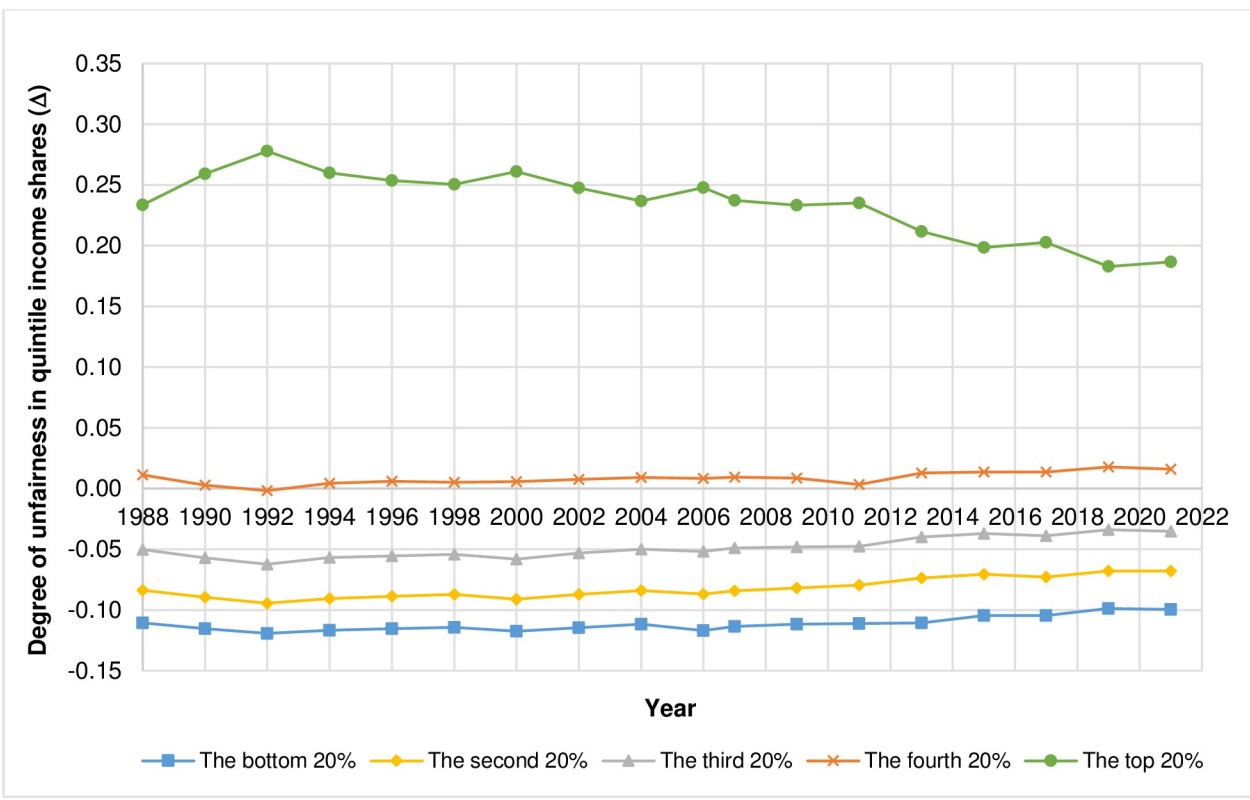

**Fig 5. The degree of unfairness in quintile income shares of Thailand from 1988 to 2021 as measured by the difference between actual quintile income shares and optimal (fair) quintile income shares (Δ) based on PK method.** Note that the scale of Δ is between– 0.15 and 0.35.

decreasing as shown by the values of Δs that are decreasing while the degree of unfairness in quintile income share of the fourth 20% is slightly increasing as shown by the increasing value of Δ.

### Fair income distribution based on SH method and optimal (fair) income distribution based on PK method as a practical guideline for designing income redistributive policies

In addition to utilizing SH method for quantifying whether income distribution in Thailand is fair, the fairness benchmarks can be used to provide a practical guideline as to whether what fair quintile income shares in Thailand should be for a particular value of income Gini index. As shown in Table 3, this can be demonstrated by using the real case study of Thailand whose one of policy targets as stated in the Twelfth National Economic and Social Development Plan is to reduce income inequality across socio-economic groups by lowering the income Gini index to 0.410 [10].

According to the NESDC [3], in 2015, Thailand has the income Gini index of 0.445 and the income shares held by the bottom 20%, the second 20%, the third 20%, the fourth 20%, and the top 20% are equal to 4.9%, 9.4%, 13.9%, 21.0%, and 50.7%, respectively. Given that there are infinite combinations of quintile income shares that can have the same value of income Gini index of 0.410 but only one of them is regarded as fair, SH method suggests that fair quintile income shares in Thailand being consistent with the targeted income Gini index of 0.410

**Table 3. Actual quintile income shares of Thailand in 2015, fair quintile income shares being consistent with the targeted income Gini index of 0.410, and optimal (fair) quintile income shares representing feasible income equality with the income Gini index of 0.137.**

| Income share | Bottom 20% | Second 20% | Third 20% | Fourth 20% | Top 20% | Income Gini index |
|---|---|---|---|---|---|---|
| Actual | 0.049 | 0.094 | 0.139 | 0.210 | 0.507 | 0.445 |
| Fair | 0.043 | 0.100 | 0.164 | 0.261 | 0.432 | 0.410 |
| Optimal | 0.154 | 0.165 | 0.176 | 0.196 | 0.309 | 0.137 |

should be such that the bottom 20% and the second 20% would receive income shares around 4.30% and 10.0%, respectively. The income share of the third 20% should be about 16.4% while the income share of the fourth 20% should be about 26.1%. The top 20% would receive income share around 43.2%.

The use of PK method as a practical guideline for designing income redistributive policies can also be demonstrated as shown in Table 3. Given the actual data on income distribution in Thailand in 2015 where the income shares of the bottom 20%, the second 20%, the third 20%, the fourth 20%, and the top 20% are equal 4.9%, 9.4%, 13.9%, 21.0%, and 50.7%, respectively, PK method indicates that, in order to achieve the optimal (fair) income distributions that characterize feasible income equality, the bottom 20% should have income share around 15.4% while the second 20% should have income share around 16.5%. The third 20% and the fourth 20% should have income shares about 17.6% and 19.6%, respectively whereas the top 20% should have income share around 30.9%. This would result in the value of income Gini index to be 0.137.

## Discussion

While reducing income inequality is considered by many as one of the top priorities for Thailand [1, 2, 7, 9–11], a thorough research by Starmans et al. [12] shows that, contrary to appearance, people around the world are not troubled by income inequality for its own sake and, indeed, they often prefer unequal income distributions both in laboratory conditions and in the real world. According to Starmans et al. [12], what really troubles people about the world we live in today is unfairness in income distribution. Therefore, it is important to investigate whether income inequality in Thailand during the past three decades is fair. If not, what fair income distribution in Thailand should be. The findings from this study could enhance the understanding of income inequality and unfair income distribution which could be useful for policymakers not only in assessing the effectiveness of income redistributive measures but also in designing policies aimed to achieve fair income distribution across all population groups in Thailand.

To examine whether inequality in income distribution in Thailand is fair and to provide a guideline on what fair inequality in income distribution in Thailand should be, this study employs the fairness benchmarks based on SH method that are derived from the distributions of salaries of athletes from various professional sports which satisfy the notions of distributive justice and procedural justice, the no-envy principle of fair allocation [26], and the general consensus or the international norm criterion of a meaningful benchmark [27]. Note again that the use of fairness benchmarks assumes that the distributions of professional athletes' salaries are stable across time. The results show that, throughout the period from 1988 to 2021, the Thai income earners in the bottom 20%, the second 20%, and the top 20% receive income shares more than the fair shares, with those in the top 20% receive income share more than the fair share the most, while the Thai income earners in the third 20% and the fourth 20% receive income shares less than the fair shares during the same period, with those in the fourth 20% receive income share less than the fair share the most. These results are, by and large, in line

with those reported in Sitthiyot and Holasut [19] who use the same fairness benchmarks to measure fair income distributions of 75 countries. They are also well supported by the empirical findings around the world in that the income growths of income earners in the third 20% and the fourth 20% are lower than those of income earners in the bottom 20%, the second 20%, and the top 20% [29–31]. According to the Organization for Economic Co-operation and Development [31], this has fueled perceptions that the current economic system is *unfair* [emphasis in original] and that the middle class has not benefited from economic growth in proportion to its contribution.

In addition to SH method, this study conducts a comparative analysis of fair income distribution in Thailand by using PK method in order to estimate optimal (fair) income distributions representing feasible income equality. The results indicate that the Thai income earners in the bottom 20%, the second 20%, and the third 20% receive income shares less than the optimal (fair) shares, with those in the bottom 20% receive income share less than the optimal (fair) share the most, whereas the Thai income earners in the fourth 20% and the top 20% receive income shares more than the optimal (fair) shares, with those in the top 20% receive income share more than the optimal (fair) share the most. These results are generally similar to those of four countries, namely, U.S.A, China, Finland, and South Africa reported in Park and Kim [18].

This study would like to note that the reason that the results on fair income shares based on SH method are different from the results on optimal (fair) income shares based on PK method is mainly because Sitthiyot and Holasut [19] and Park and Kim [18] use different concepts of fairness in income distribution as a basis for deriving their methods. According to Park and Kim [18], income distribution is regarded as fair if income shares that individuals receive are equal. However, recognizing that perfect equality in income distribution is idealistic and infeasible in the real world, Park and Kim [18] estimate optimal (fair) income distributions that represent feasible income equality. Park and Kim [18] view that, in a feasible income equality society, income should be fairly distributed among individuals.

For Sitthiyot and Holasut [19], income distribution is viewed as fair if it satisfies the notions of distributive justice and procedural justice. Based on these two notions, Sitthiyot and Holasut [19] derive the fairness benchmarks for measuring whether income distribution is fair for a particular value of the income Gini index. Given that the value of income Gini index is between 0 and 1, where 0 means perfect income equality and 1 means perfect income inequality, a fair income distribution society, according to Sitthiyot and Holasut [19], does not necessarily have to be a society whose population have more or less equal income with the income Gini index being close to 0 as viewed by Park and Kim [18]. Rather, a society can choose any value of income Gini index that it would like to achieve, provided that the chosen value of income Gini index is more than 0 but less than 1, and SH method would provide the information on what fair income distribution, being consistent with the chosen income Gini index, should be.

Furthermore, this study demonstrates the use of SH method and PK method as a practical guideline for designing income redistributive policies by using actual income distribution of Thailand in 2015 as a case study where the income shares of the bottom 20%, the second 20%, the third 20%, the fourth 20%, and the top 20% are equal to 4.9%, 9.4%, 13.9%, 21.0%, and 50.7%, respectively, and the income Gini index is equal to 0.445 [10]. Given that SH method and PK method use different concepts of fair income distribution as discussed above, the use of SH method and PK method as a practical guideline for designing income redistributive policies would depend upon the aim that policymakers would like to achieve.

If the aim is to achieve the income Gini index of 0.410 as stated in Thailand Twelfth National Economic and Social Development Plan [10], policymakers could employ SH method which indicates that the fair quintile income shares for the bottom 20%, the second 20%, the third 20%, the fourth 20%, and the top 20% should be 4.3%, 10.0%, 16.4%, 26.1%, and

43.2%, respectively. Policymakers could use this information for designing policies in order to not only reduce income inequality but also achieve fair income distribution at the same time by redistributing income from the top 20% to the third 20% and the fourth 20% while the income shares of the bottom 20% and the second 20% remain mostly unaffected.

However, if the aim is to achieve feasible income equality, policymakers could use PK method which suggests that the optimal (fair) income shares of the bottom 20%, the second 20%, the third 20%, the fourth 20%, and the top 20% should equal 15.4%, 16.5%, 17.6%, 19.6%, and 30.9%, respectively, resulting in the value of the income Gini index to be 0.137. The results based on PK method indicate that, in order to achieve feasible income equality, policymakers should design policies in order to redistribute income from the top 20% to the bottom 80%. Fig 6 shows the Lorenz plot of actual income distribution by quintile of Thailand in 2015 corresponding to the income Gini index of 0.445, the Lorenz plot of fair income distribution by quintile based on SH method that is consistent with the targeted income Gini index of 0.410, and the Lorenz plot of optimal (fair) income distribution representing feasible income equality based on PK method with the income Gini index of 0.137.

While the main concern about income inequality in Thailand, besides high income inequality, is the income earners who are in the bottom 40% [10], the overall results from this study

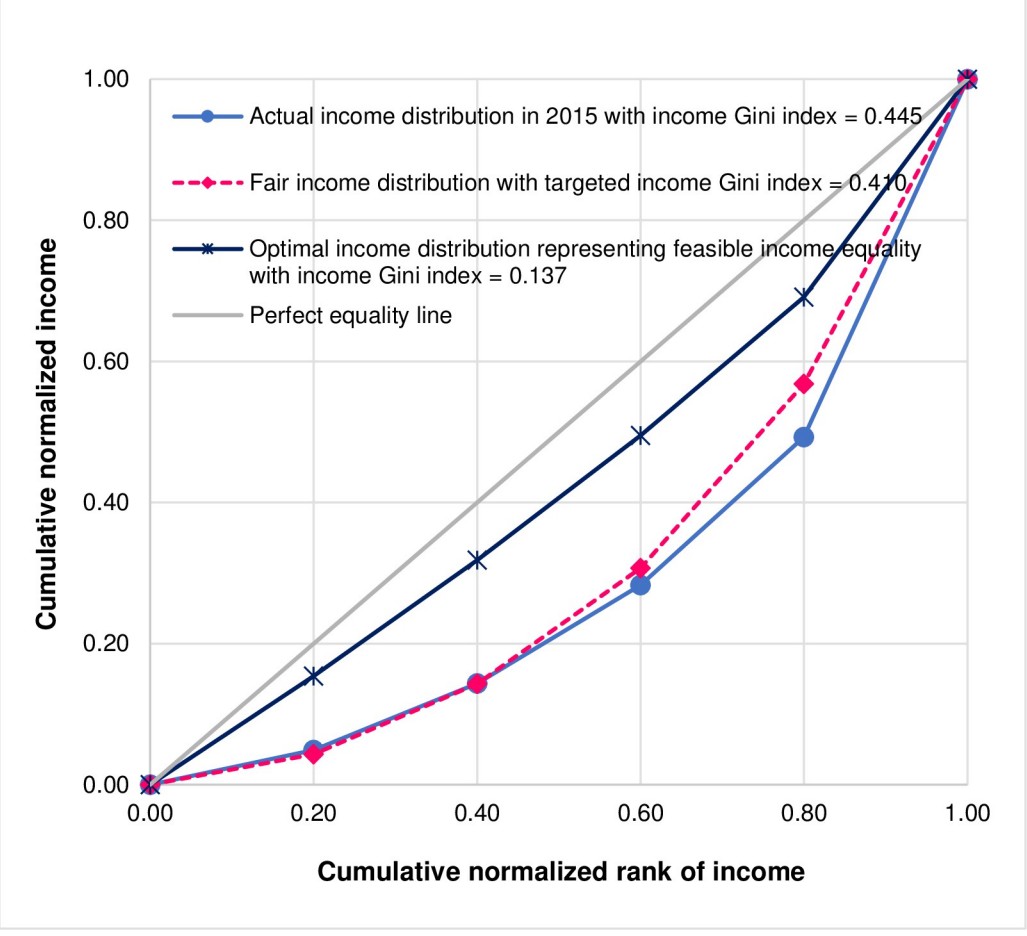

**Fig 6. Lorenz plot of actual income distribution by quintile of Thailand in 2015 corresponding to the income Gini index of 0.445, Lorenz plot of fair income distribution by quintile based on SH method that is consistent with the targeted Gini index of 0.410, and Lorenz plot of optimal (fair) income distribution by quintile representing feasible income equality based on PK method with the Gini index of 0.137.**

suggest that, instead of conducting income redistributive policies that target the low-income population, namely the bottom 40%, with the aim to reduce income inequality in Thailand [10], policymakers should put more emphasis on policies that would result in fair income distribution across *all* groups of population. This is in line with the "Leave No One Behind Principle" which is the central and transformative promise of the 2030 Agenda for Sustainable Development and its SDGs [32]. The challenging issue is how to design and implement income distribution policies on the bases of procedural justice and distributive justice similar to those in professional sports such that the Thai population across socio-economic groups would unconditionally and whole-heartedly perceive as fair.

## Supporting information

**S1 Table. The calculated values of L, H, μ and α as well as the estimated values of W and β\* based on PK method.**
(PDF)

## Acknowledgments

The authors sincerely thanks Dr. Suradit Holasut and five Reviewers for guidance and comments.

## Author Contributions

**Conceptualization:** Thitithep Sitthiyot.

**Formal analysis:** Thitithep Sitthiyot.

**Funding acquisition:** Thitithep Sitthiyot.

**Methodology:** Thitithep Sitthiyot, Kanyarat Holasut.

**Project administration:** Thitithep Sitthiyot.

**Validation:** Kanyarat Holasut.

**Writing – original draft:** Thitithep Sitthiyot.

**Writing – review & editing:** Thitithep Sitthiyot, Kanyarat Holasut.

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
