## [Decision Letter · Decision Letter 0]

20 Nov 2023

PONE-D-23-25866Quantifying fair income distribution in ThailandPLOS ONE

Dear Dr. Thititheo Sitthiyot,

Thank you for submitting your manuscript to PLOS ONE. After careful consideration, we feel that it has merit but does not fully meet PLOS ONE’s publication criteria as it currently stands. Therefore, we invite you to submit a revised version of the manuscript that addresses the points raised during the review process.

Specifically:Give explanation of how eq.(1)-5 were derivedStrengthen  the paper's contribution by conducting robustness checksExplain the contribution of your paper to the extant literature

We look forward to receiving your revised manuscript.

Kind regards,

Stephen Esaku

Academic Editor

PLOS ONE

Journal Requirements:

Chapter 4: Technology and Inequalities. Inequality in Asia and the Pacific in the Era of the 2030 Agenda for Sustainable Development - https://www.researchgate.net/publication/354044970_Chapter_4_Technology_and_Inequalities_Inequality_in_Asia_and_the_Pacific_in_the_Era_of_the_2030_Agenda_for_Sustainable_Development

In your revision ensure you cite all your sources (including your own works), and quote or rephrase any duplicated text outside the methods section. Further consideration is dependent on these concerns being addressed.

Reviewers' comments:

Reviewer's Responses to Questions

**Comments to the Author**

1. Is the manuscript technically sound, and do the data support the conclusions?

Reviewer #1: Partly

Reviewer #2: No

2. Has the statistical analysis been performed appropriately and rigorously? 

Reviewer #1: No

Reviewer #2: No

3. Have the authors made all data underlying the findings in their manuscript fully available?

Reviewer #1: Yes

Reviewer #2: No

4. Is the manuscript presented in an intelligible fashion and written in standard English?

Reviewer #1: Yes

Reviewer #2: Yes

5. Review Comments to the Author

Reviewer #1: This study examines income inequality in Thailand over the past three decades and determines whether it is fair. It also defines what fair inequality should be in Thailand. The study uses fairness benchmarks based on distributive and procedural justice to quantify fair inequality. The results show that the bottom 20%, second 20%, and top 20% of income earners receive more than their fair share, while the third 20% and fourth 20% receive less than their fair share. These findings can be used to evaluate the effectiveness of income redistribution measures and to formulate policies aimed at achieving fair income distribution in Thailand. The study also demonstrates that fairness benchmarks can be used as a practical guideline for designing income redistribution policies and measures. Below I have some suggestions.

Regarding Fig. 2, it would be beneficial for the authors to provide a more explicit rationale for their interpretation of the scatter plots. The current presentation lacks clarity on why the data align with the distribution or fit lines, and additional details would enhance the persuasiveness of this analysis.

I recommend conducting a separation analysis, specifically comparing the top 20% and bottom 20% data distributions. This would offer valuable insights into potential variations in fairness and income across different groups. Highlighting the distinctions between these two extremes could contribute significantly to the study's comprehensiveness.

It is essential to articulate the significance of this study. Why is it important, and what broader implications does it hold? Additionally, consider discussing the fairness of income distribution in the context of other countries and exploring any relationships that may emerge. Providing this contextualization will enrich the paper and strengthen its contribution to the field.

Reviewer #2: This paper contributes to the literature on fair income distribution by empirically applying the method developed by Sitthiyot and Holasut (2022) in Thailand. While the analysis of alternative measures to evaluate inequality levels is crucial for understanding the magnitude of unfair inequalities, the paper has several drawbacks that need attention before I can recommend it for publication. In the following sections, I will outline the main limitations of the study in the hope that these points will assist the authors in enhancing their analysis.

(1) There exists a vast literature on unfair inequality, and I recommend the inclusion of additional references to provide a more comprehensive theoretical and normative framework for measuring the unfair distribution of income. Notable works in this field include Almås et al (2011), Cappelen, A. W., & Tungodden (2017), and Hufe et al. (2022). While the authors acknowledge contributions from the econophysics stream, it is essential to recognize the more traditional economic literature, which has also significantly contributed to the understanding of income distribution.

(2) The authors chose to exclusively apply the method they developed in a previous paper (Sitthiyot and Holasut, 2022). While this choice is valid, it is crucial to complement this analysis with other approaches used by previous studies in order to validate the robustness of the conclusions. A comparative analysis of different methods would strengthen the paper's contribution to the field.

(3) The paper lacks an explanation of how Equations (1) – (5) are derived. Although it is assumed they originate from Sitthiyot and Holasut (2022), no reference to this is made in the text. The authors should clarify how these equations are obtained, the data used for estimation, and how this data compares with the dataset used in the empirical application developed in this paper.

(4) Considering the period covered (1988 – 2021), it is questionable whether Eqs. (1) to (5) can accurately measure the fair distribution over this extended timeframe. Changes in the income distribution of athletes, considered the fair distribution in this context, may have occurred over the last 30 years. The analysis should account for these potential changes to enhance the temporal validity of the study.

(5) The main conclusion asserting that the bottom 40 per cent owns more income than what would be deemed fair is a strong statement. Given the potential policy implications, such as redistributing income from these quantiles to the top, the authors should address the concern that this could exacerbate poverty levels in a country where 6.3 per cent of the population is already classified as poor. A more nuanced discussion of the potential consequences is warranted.

References

Almås, I., Cappelen, A. W., Lind, J. T., Sørensen, E. Ø., & Tungodden, B. (2011). Measuring unfair (in) equality. Journal of Public Economics, 95(7-8), 488-499.

Cappelen, A. W., & Tungodden, B. (2017). Fairness and the proportionality principle. Social Choice and Welfare, 49, 709-719

Hufe, P., Kanbur, R., & Peichl, A. (2022). Measuring unfair inequality: Reconciling equality of opportunity and freedom from poverty. The Review of Economic Studies, 89(6), 3345-3380.

Sitthiyot T, Holasut K. 2022. A quantitative method for benchmarking fair income distribution. Heliyon. 8, e10511. https://doi.org/10.1016/j.heliyon.2022.e10511

6. PLOS authors have the option to publish the peer review history of their article (what does this mean?). If published, this will include your full peer review and any attached files.

Reviewer #1: No

Reviewer #2: No

---

## [Author Response · Author response to Decision Letter 0]

22 Jan 2024

Response to Academic Editor

First of all, we would like to sincerely thank you for giving us an opportunity to revise our manuscript. We are re-submitting the revised version of our original research paper entitled “Quantifying fair income distribution in Thailand” for your consideration for publication in PLOS ONE. We would like to inform you that we have made a substantial revision by incorporating comments, concerns, and suggestions made by Reviewers #1 and #2 into our revised manuscript. We also provide explanation of how Equations (1) – (5) are derived, conduct a comparative analysis by using an alternative method for estimating optimal (fair) income distribution representing feasible income equality developed by Park and Kim (2021) to strengthen our paper, and explain the contribution of our paper to the existing literature as you specifically suggest. All adjustments and changes are highlighted in yellow. 

If there is anything else that we could do to improve the quality of our revised manuscript in order to bring it up to a publishable level in PLOS ONE, please let us know. Again, thank you very much for giving us the opportunity to revise our manuscript. We look forward to hearing feedback from you in the near future.

Response to Reviewers

We sincerely thank Reviewer #1 and Reviewer #2 for providing several important comments, concerns, and suggestions. We would like to inform both Reviewers that we have made a substantial revision according to comments, concerns, and suggestions made by both Reviewers. We hope that our revised manuscript is clearer in all aspects that Reviewer #1 and Reviewer #2 have commented, concerned, and suggested. Let us respond to both Reviewers’ comments, concerns, and suggestions as follows.

Reviewer #1

This study examines income inequality in Thailand over the past three decades and determines whether it is fair. It also defines what fair inequality should be in Thailand. The study uses fairness benchmarks based on distributive and procedural justice to quantify fair inequality. The results show that the bottom 20%, second 20%, and top 20% of income earners receive more than their fair share, while the third 20% and fourth 20% receive less than their fair share. These findings can be used to evaluate the effectiveness of income redistribution measures and to formulate policies aimed at achieving fair income distribution in Thailand. The study also demonstrates that fairness benchmarks can be used as a practical guideline for designing income redistribution policies and measures. Below I have some suggestions.

- Regarding Fig. 2, it would be beneficial for the authors to provide a more explicit rationale for their interpretation of the scatter plots. The current presentation lacks clarity on why the data align with the distribution or fit lines, and additional details would enhance the persuasiveness of this analysis.

In response to Reviewer #1’s comment with regard to the interpretation of results as shown in Fig. 2, we would like to inform Reviewer #1 that we explain the rationale for the interpretation of scatter plots of actual quintile income shares relative to their corresponding fairness benchmarks in Materials and Methods, paragraphs 5 and 6 which we use it as a basis for interpreting our results as provided in Results, paragraphs 2, 3, and 4 in our revised manuscript.

- I recommend conducting a separation analysis, specifically comparing the top 20% and bottom 20% data distributions. This would offer valuable insights into potential variations in fairness and income across different groups. Highlighting the distinctions between these two extremes could contribute significantly to the study's comprehensiveness.

We would like to thank Reviewer #1 for this useful suggestion. We add the discussion on variations in unfairness in income distribution across different population groups in Results, paragraphs 5 and 8 in our revised manuscript.

- It is essential to articulate the significance of this study. Why is it important, and what broader implications does it hold? Additionally, consider discussing the fairness of income distribution in the context of other countries and exploring any relationships that may emerge. Providing this contextualization will enrich the paper and strengthen its contribution to the field.

In response to Reviewer #1’s suggestion on articulating the significance of our study and its implications, we state the significance of our study in Introduction, paragraphs 3 and 9, and in Discussion, paragraph 1 in our revised manuscript.

The implications of our study are also provided in Results, paragraphs 9 – 11 and in Discussion, paragraphs 6 – 8 in our revised manuscript.

In addition, we discuss the fairness in income distribution in the context of other countries and explore the similarity and/or difference among them in Discussion, paragraphs 2 and 3 in our revised manuscript as suggested by Reviewer #1.

Reviewer #2

This paper contributes to the literature on fair income distribution by empirically applying the method developed by Sitthiyot and Holasut (2022) in Thailand. While the analysis of alternative measures to evaluate inequality levels is crucial for understanding the magnitude of unfair inequalities, the paper has several drawbacks that need attention before I can recommend it for publication. In the following sections, I will outline the main limitations of the study in the hope that these points will assist the authors in enhancing their analysis.

(1) There exists a vast literature on unfair inequality, and I recommend the inclusion of additional references to provide a more comprehensive theoretical and normative framework for measuring the unfair distribution of income. Notable works in this field include Almås et al (2011), Cappelen, A. W., & Tungodden (2017), and Hufe et al. (2022). While the authors acknowledge contributions from the econophysics stream, it is essential to recognize the more traditional economic literature, which has also significantly contributed to the understanding of income distribution.

We sincerely thank Reviewer #2 for this suggestion. We would like to inform Reviewer #2 that we include studies by Almås et al. (2011), Cappelen and Tungodden (2017), and Hufe et al. (2022) in Introduction, paragraph 3 in our revised manuscript. We do learn a lot from these studies.

(2) The authors chose to exclusively apply the method they developed in a previous paper (Sitthiyot and Holasut, 2022). While this choice is valid, it is crucial to complement this analysis with other approaches used by previous studies in order to validate the robustness of the conclusions. A comparative analysis of different methods would strengthen the paper's contribution to the field.

We would like to thank Reviewer #2 for this suggestion, in our revised manuscript, we conduct a comparative analysis by using the method for estimating optimal income distribution representing feasible fair equality developed by Park and Kim (2021) in order to provide an alternative recommendation for designing policies aimed to achieve fair income distribution. The main reason for choosing Park and Kim (2021)’s method is due to the availability of data of Thailand that can be used to conduct the analysis. We would like to inform Reviewer #2 that we report the calculated and estimated values of all parameters based on Park and Kim (2021)’s method in S1 Table in Supporting Information.

(3) The paper lacks an explanation of how Equations (1) – (5) are derived. Although it is assumed they originate from Sitthiyot and Holasut (2022), no reference to this is made in the text. The authors should clarify how these equations are obtained, the data used for estimation, and how this data compares with the dataset used in the empirical application developed in this paper.

In response to Reviewer #2 on this issue, we would like to inform Reviewer #2 that we provide the derivation of Equations (1) – (5) and the data used for estimation in Materials and Methods, paragraphs 1 – 4 in our revised manuscript. We also cite our work (Sitthiyot and Holasut, 2022) in the text and provide reference in our revised manuscript. In addition, the dataset used for empirical analysis is stated in Materials and Methods, paragraph 12 and in References in our revised manuscript.

(4) Considering the period covered (1988 – 2021), it is questionable whether Eqs. (1) to (5) can accurately measure the fair distribution over this extended timeframe. Changes in the income distribution of athletes, considered the fair distribution in this context, may have occurred over the last 30 years. The analysis should account for these potential changes to enhance the temporal validity of the study.

We totally agree with Reviewer #2 on this issue and would like to thank Reviewer #2 for pointing this out. For Eqs. (1) – (5) to be used for measuring fair income distribution, we make an assumption that the distributions of professional athletes’ salaries are stable across time. We state this in Materials and Methods, paragraph 4 and in Discussion, paragraph 2 in our revised manuscript. 

(5) The main conclusion asserting that the bottom 40 per cent owns more income than what would be deemed fair is a strong statement. Given the potential policy implications, such as redistributing income from these quantiles to the top, the authors should address the concern that this could exacerbate poverty levels in a country where 6.3 per cent of the population is already classified as poor. A more nuanced discussion of the potential consequences is warranted.

We would like to sincerely thank Reviewer #2 for addressing this important point. In response to concern and suggestion made by Reviewer #2 on this issue, we would like to inform Reviewer #2 that we have made a major adjustment in our revised manuscript by seriously taking Reviewer #2’s concern and suggestion into account and rewriting policy implication based on our results regarding the bottom 40% by using the actual quintile income shares and the income Gini index of Thailand in 2015 as a case study.

Given the goal is to achieve the income Gini index of 0.410 as specified in Thailand’s Twelfth National Economic and Social Development Plan, policymakers could use the information on fair quintile income shares based on the fairness benchmarks for designing policy in order to not only lower income inequality but also achieve fair income distribution across all population groups at the same time by redistributing income from the top 20% to the third 20% and the fourth 20% while the income shares of the bottom 20% and the second 20% are mostly unaffected. We discuss all of these in Discussion, paragraphs 6 and 7 in our revised manuscript and also use Fig. 6 to supplement our discussion on this issue.

---

## [Decision Letter · Decision Letter 1]

7 Feb 2024

PONE-D-23-25866R1Quantifying fair income distribution in ThailandPLOS ONE

Dear Dr. Sitthiyot,

Thank you for submitting your manuscript to PLOS ONE. After careful consideration, we feel that it has merit but does not fully meet PLOS ONE’s publication criteria as it currently stands. Therefore, we invite you to submit a revised version of the manuscript that addresses the points raised during the review process.

Specifically:**Improve on the literature review****Address all comments raised by reviewers**Please submit your revised manuscript by Mar 23 2024 11:59PM. If you will need more time than this to complete your revisions, please reply to this message or contact the journal office at plosone@plos.org. Please include the following items when submitting your revised manuscript:A rebuttal letter that responds to each point raised by the academic editor and reviewer(s). You should upload this letter as a separate file labeled 'Response to Reviewers'.A marked-up copy of your manuscript that highlights changes made to the original version. You should upload this as a separate file labeled 'Revised Manuscript with Track Changes'.An unmarked version of your revised paper without tracked changes. You should upload this as a separate file labeled 'Manuscript'.

We look forward to receiving your revised manuscript.

Kind regards,

Stephen Esaku

Academic Editor

PLOS ONE

Reviewers' comments:

Reviewer's Responses to Questions

**Comments to the Author**

1. If the authors have adequately addressed your comments raised in a previous round of review and you feel that this manuscript is now acceptable for publication, you may indicate that here to bypass the “Comments to the Author” section, enter your conflict of interest statement in the “Confidential to Editor” section, and submit your "Accept" recommendation.

Reviewer #3: All comments have been addressed

Reviewer #4: All comments have been addressed

Reviewer #5: All comments have been addressed

2. Is the manuscript technically sound, and do the data support the conclusions?

Reviewer #3: No

Reviewer #4: No

Reviewer #5: Yes

3. Has the statistical analysis been performed appropriately and rigorously? 

Reviewer #3: Yes

Reviewer #4: I Don't Know

Reviewer #5: Yes

4. Have the authors made all data underlying the findings in their manuscript fully available?

Reviewer #3: No

Reviewer #4: No

Reviewer #5: Yes

5. Is the manuscript presented in an intelligible fashion and written in standard English?

Reviewer #3: No

Reviewer #4: Yes

Reviewer #5: Yes

6. Review Comments to the Author

Reviewer #3: 1. It is necessary to refine the abstract. In the current state the paper, due to the language it is very challenging to understand the narrative. It is need to specify: the initial data, the hypothesis of the study to strengthen, specify the scope of application, limitations/directions of future research. For example, How to understand this "Given growing concern about high income inequality in Thailand as opposed to empirical findings around the world showing people’s preference for fair income inequality over unfair income equality, it is therefore important to examine whether inequality in income distribution in Thailand over the past three decades is fair, and what fair inequality in income distribution in Thailand should be"?

2. I believe there are findings from the context of your study that will have interest for scholars and policy makers.However, the purpose of the study is formulated very incorrectly and it should be refined.

3. The introduction should outline the context of the problem and justify the need for research (research gap, a new phenomenon).

4. Unfortunately, I did not understand the literature review.Literature review is also completely chaotic. Why is it dedicated? It should be fully subordinated to the coverage of the results of previous scientific research on the subject, which will be determined in the purpose of your research.

5. The methods should be described as concretely as possible. The section may include a description of data, methods of collecting, etc.

Reviewer #4: I was not one of the first-round reviewers so this was the first time I read the paper. I acknowledge the professional and rigorous approach you have taken to answering the previous reviewers. I really appreciate the opportunity to read this paper and I can see there are some valuable ideas within it. Unfortunately, I am unable to recommend acceptance and I do not think even a thoroughly revised version of this paper would be suitable for PLOS-ONE.

I enjoyed trying to get to grips with the intuition underlying this paper. What you have argued in previous work is that for whatever GINI is deemed acceptable, there is a curve that selects the approximate distribution between income quintiles that would emerge from competition based on fair and transparent rules, using professional sports rules and their resulting income distributions as the baseline for this judgement. I think there is a lot that could be said to challenge this baseline but it is clever and aligns with the way people are generally unconcerned with inequality that emerges between sports people competing under the same rules.

In this paper, you apply this to evaluating the income distribution of Thailand and show how the distribution between quintiles would have to change whether following your baseline judgement or Park and Kim’s (2021) more demanding approach to tackling inequality.

The key question is how this develops your previous work as, on its face, ‘A quantitative method for benchmarking fair income distribution’ makes exactly the same point but, more ambitiously, for 75 countries. I cannot detect a compelling answer. The new paper’s contribution is to consider Thailand’s GINI over 30 years rather than a single year snapshot. That is potentially very interesting but the paper discusses very little of the changing economic structure or policy over that thirty year period. In terms of analysis, it might be as well be 30 (pretty similar) statistical snapshots, each independently judged against two normative baselines, rather than a narrative that might indicate what sort of policies might make an income distribution fairer. What sort of rules and economic institutions are Thai people subject to and how do they differ from the fair and transparent rules of professional sports? We learn very little about this.

Critically, one could imagine policies that arrange income into the appropriate quintiles but without being fair and transparent. So, the outcomes might look ‘fair’ but they would not be fair as people experience them. I reminded of points made by Acemoglu and Robinson reviewing Piketty where they point out the changing distributions of South Africa and Sweden look similar but one is based on exploitative institutions whereas the others are remarkably fair. There is not more going on to judge a process than a particular shape of the outcomes:

Acemoglu, D., & Robinson, J. (2015). The Rise and Decline of General Laws of Capitalism. Journal of Economic Perspectives, 29(1), 3–28.

On this account, if the proposed distribution were made a simple ‘target’ to hit, policymakers could still easily fail to create outcomes that people within the ‘game’ policymakers have set up would judge as fair. So overall the addition of the temporal element does not seem to have expanded the analysis in a compelling or insightful on top of what you have already done.

I have a more prosaic concern with regards to the data. In your previous paper (Appendix Table A1), Thailand is listed as having a GINI index of 0.360. This is pretty close to the World Bank Databank’s most recent estimate of 0.351 so it seems to be correct in that paper. That also chimes with the Asian Development Bank’s narrative:

‘Thailand’s GINI coefficient on the expenditure side fell from a peak of 0.48 in 1992 to 0.35 in 2019.’

https://www.adb.org/sites/default/files/linked-documents/tha-cps-2021-2025-ld01.pdf

Yet, in the new paper the GINI is always 0.43 or above. The paper specifies that this is the income GINI (does not discuss expenditure explicitly) but does this suggest that the expenditure GINI used in the previous paper (and elsewhere) was a misapplication? Why the shift to a different series for this paper? This matters since the premise for the focus on Thailand is that it is remarkably unequal compared to other countries in the region. But according to the WorldBank, it is not, or at least not so much anymore. We would need to see income GINIs for the other countries to know this was a case worth exploring the in the first instance.

Reviewer #5: This manuscript presents a new methodology for assessing the fairness of income distribution in Thailand. It utilizes the concepts of distributive and procedural justice to quantify what is considered fair inequality, analyzing whether specific income brackets are receiving more or less than their fair share. This research could aid in evaluating the effectiveness of income redistribution policies and in formulating strategies for achieving fair income distribution.

Following the suggestions of Reviewers 1 and 2, the authors have made significant improvements to the paper. These enhancements include clarifying the study's importance and implications, as well as providing detailed explanations of the research methodology and data.

Additional Reviewer Comments:

However, the following further improvements are necessary:

Clear Definition of Distributive Justice Required: The paper mentions distributive justice but lacks a specific definition or theoretical basis. Since various definitions and interpretations of distributive justice exist, the authors must clarify which definition they are using and the rationale behind this choice. This clarification is important for understanding the research methodology and conclusions.

Improvement in Graphics Resolution: The resolution of graphs needs to be enhanced to more clearly convey the data and analysis results. This improvement is vital for effectively communicating the research findings.

The methodology and results proposed in this paper have the potential to enhance the understanding of fair income distribution and could provide valuable insights to academia and policy-makers. Considering the above aspects, this manuscript appears to have sufficient potential for publication in PLOS ONE.

7. PLOS authors have the option to publish the peer review history of their article (what does this mean?). If published, this will include your full peer review and any attached files.

Reviewer #3: No

Reviewer #4: **Yes: **Nick Cowen

Reviewer #5: No

---

## [Author Response · Author response to Decision Letter 1]

28 Feb 2024

Dear Reviewers,

We sincerely thank all three Reviewers for providing a number of important comments and making clear suggestions with the view of improving the quality and clarity of our paper. We have addressed all of the Reviewers’ and the Academic Editor’s comments and tried our best to incorporate all comments and suggestions into our revised manuscript. We hope that our revised paper is clearer in all aspects that the Reviewers and the Academic Editor have commented and suggested. Let us respond to each Reviewer’s and the Academic Editor’s comments and suggestions as follows.

Reviewer #3 

1. It is necessary to refine the abstract. In the current state the paper, due to the language it is very challenging to understand the narrative. It is need to specify: the initial data, the hypothesis of the study to strengthen, specify the scope of application, limitations/directions of future research. For example, How to understand this "Given growing concern about high income inequality in Thailand as opposed to empirical findings around the world showing people’s preference for fair income inequality over unfair income equality, it is therefore important to examine whether inequality in income distribution in Thailand over the past three decades is fair, and what fair inequality in income distribution in Thailand should be"?

We sincerely thank Reviewer #3 for this comment. In our revised manuscript, we rewrite Abstract as suggested by Reviewer #3.

2. I believe there are findings from the context of your study that will have interest for scholars and policy makers. However, the purpose of the study is formulated very incorrectly and it should be refined.

We would like to sincerely thank Reviewer #3 for finding our paper useful. In our revised manuscript, we rewrite the purpose of the study in Abstract, Introduction, paragraph 3, and Discussion, paragraph 1.

3. The introduction should outline the context of the problem and justify the need for research (research gap, a new phenomenon).

In response to Reviewer #3 on this issue, we have made a substantial revision by rewriting the whole Introduction section in our revised manuscript.

4. Unfortunately, I did not understand the literature review. Literature review is also completely chaotic. Why is it dedicated? It should be fully subordinated to the coverage of the results of previous scientific research on the subject, which will be determined in the purpose of your research.

We would like to sincerely thank Reviewer #3 for this comment. In Introduction, paragraphs 6 and 8-12, in our revised manuscript, we have made a major revision by rewriting the literature review as suggested by Reviewer #3 and the Academic Editor.

5. The methods should be described as concretely as possible. The section may include a description of data, methods of collecting, etc.

In response to Reviewer #3 on this issue, we would like to inform Reviewer #3 that the previous version of our method was shorter than the current one. The current version is the product based on comments made by Reviewer #2 who, in the first round of review, asked us to explain in details how equations for fairness benchmarks are derived and also to conduct a comparative analysis by using different methods of measuring fair income distribution. In addition, we would like to inform Reviewer #3 that the description of data used in our study and how we obtained them are described in Methods, paragraph 12 and in References in our revised manuscript.

Reviewer #4 

- I was not one of the first-round reviewers so this was the first time I read the paper. I acknowledge the professional and rigorous approach you have taken to answering the previous reviewers. I really appreciate the opportunity to read this paper and I can see there are some valuable ideas within it. Unfortunately, I am unable to recommend acceptance and I do not think even a thoroughly revised version of this paper would be suitable for PLOS-ONE.

I enjoyed trying to get to grips with the intuition underlying this paper. What you have argued in previous work is that for whatever GINI is deemed acceptable, there is a curve that selects the approximate distribution between income quintiles that would emerge from competition based on fair and transparent rules, using professional sports rules and their resulting income distributions as the baseline for this judgement. I think there is a lot that could be said to challenge this baseline but it is clever and aligns with the way people are generally unconcerned with inequality that emerges between sports people competing under the same rules.

In this paper, you apply this to evaluating the income distribution of Thailand and show how the distribution between quintiles would have to change whether following your baseline judgement or Park and Kim’s (2021) more demanding approach to tackling inequality.

The key question is how this develops your previous work as, on its face, ‘A quantitative method for benchmarking fair income distribution’ makes exactly the same point but, more ambitiously, for 75 countries. I cannot detect a compelling answer. The new paper’s contribution is to consider Thailand’s GINI over 30 years rather than a single year snapshot. That is potentially very interesting but the paper discusses very little of the changing economic structure or policy over that thirty year period. In terms of analysis, it might be as well be 30 (pretty similar) statistical snapshots, each independently judged against two normative baselines, rather than a narrative that might indicate what sort of policies might make an income distribution fairer. What sort of rules and economic institutions are Thai people subject to and how do they differ from the fair and transparent rules of professional sports? We learn very little about this.

In response to Reviewer #4 on this issue, we would like to clarify with Reviewer #4 that the main objective of our previous paper is to introduce a new quantitative method for measuring fair income distribution. The reason that we use the data of 75 countries from the World Bank is mainly to demonstrate how our method works for various countries instead of focusing on one specific country.

For the current manuscript, we show how the method that we previously developed can be applied to analyze fair income distribution across time using Thailand as a case study. We do acknowledge the importance of rules and economic institutions the Thai people are subject to and how they differ from the fair and transparent rules of professional sports but this topic is beyond the scope of our study and worth for future investigation.

- Critically, one could imagine policies that arrange income into the appropriate quintiles but without being fair and transparent. So, the outcomes might look ‘fair’ but they would not be fair as people experience them. I reminded of points made by Acemoglu and Robinson reviewing Piketty where they point out the changing distributions of South Africa and Sweden look similar but one is based on exploitative institutions whereas the others are remarkably fair. There is not more going on to judge a process than a particular shape of the outcomes:

Acemoglu, D., & Robinson, J. (2015). The Rise and Decline of General Laws of Capitalism. Journal of Economic Perspectives, 29(1), 3–28.

On this account, if the proposed distribution were made a simple ‘target’ to hit, policymakers could still easily fail to create outcomes that people within the ‘game’ policymakers have set up would judge as fair. So overall the addition of the temporal element does not seem to have expanded the analysis in a compelling or insightful on top of what you have already done.

In response to Reviewer #4’s comment on this issue, we state in Discussion, paragraph 9 in our revised manuscript that, in order to achieve fair income distribution for a particular value of income Gini index, policymakers have to design and implement income distribution policies on the bases of procedural justice and distributive justice similar to those in professional sports such that the Thai population across socio-economic groups would unconditionally and whole-heartedly perceive as fair. 

- I have a more prosaic concern with regards to the data. In your previous paper (Appendix Table A1), Thailand is listed as having a GINI index of 0.360. This is pretty close to the World Bank Databank’s most recent estimate of 0.351 so it seems to be correct in that paper. That also chimes with the Asian Development Bank’s narrative:

‘Thailand’s GINI coefficient on the expenditure side fell from a peak of 0.48 in 1992 to 0.35 in 2019.’

https://www.adb.org/sites/default/files/linked-documents/tha-cps-2021-2025-ld01.pdf

Yet, in the new paper the GINI is always 0.43 or above. The paper specifies that this is the income GINI (does not discuss expenditure explicitly) but does this suggest that the expenditure GINI used in the previous paper (and elsewhere) was a misapplication? Why the shift to a different series for this paper? This matters since the premise for the focus on Thailand is that it is remarkably unequal compared to other countries in the region. But according to the WorldBank, it is not, or at least not so much anymore. We would need to see income GINIs for the other countries to know this was a case worth exploring the in the first instance.

We would like to sincerely thank Reviewer #4 for your comment on the use of data on Gini index from the World Bank in our previous paper. We would like to inform Reviewer #4 that we used the data on Gini index from the World Bank on the premise that the data come from the same source and they are the “World Bank estimate”. Examples of the information with regard to the data on Gini index of Thailand and other countries in Excel format that we downloaded from the World Bank website back in early 2020 are shown in the attached file (Response to Reviewers_R2).

In response to Reviewer #4’s comment on the issue whether Thailand still has high income inequality, we would like to inform Reviewer #4 that, according to the World Bank (2022) which we cite this report in Introduction, paragraph 1 in our revised manuscript, Thailand has the highest income inequality in East Asia in 2019. 

The World Bank. 2022. Thailand Rural Income Diagnostic: Challenges and Opportunities for Rural Farmers. https://www.worldbank.org/en/country/thailand/publication/thailand-rural-income-diagnostic-challenges-and-opportunities-for-rural-farmers. 

In addition to the issue whether income inequality in Thailand is still high as noted by Reviewer #4, we would like to inform Reviewer #4 that the more important issue which is the main objective of our study is to investigate whether income inequality in Thailand is fair by using the fairness benchmarks introduced in our previous paper. Even though Thailand were considered a low income inequality country, it would still be worthwhile to explore whether income distribution in Thailand is fair. This is because it is possible that a country has low income inequality but the distribution of income in that country is not fair. 

In response to Reviewer #4’s question regarding the use of data on income shares by quintile and income Gini index from the Office of National Economic and Social Development Council of Thailand (NESDC), we would like to inform Reviewer #4 that since we investigate fair income distribution in Thailand, it would be appropriate to use the data from the NESDC which is the government’s official source that provides the data on Thailand’s income statistics.

Although the NESDC website has an English language version, the income data can be accessed from the NESDC’s website in Thai language version only. The link to the NESDC website is https://www.nesdc.go.th/main.php?filename=PageSocial. If readers go to this link and click “สถิติด้านความยากจนและการกระจายรายได้” (in Thai) which can be translated as “Poverty and income distribution statistics”, readers can download the data on income shares by quintile and income Gini index of Thailand from 1988 to 2021 which are in Tables 8.1 and 8.2. The step-by-step of how to access the data on quintile income share and income Gini Index is shown in the attached file (Response to Reviewers_R2).

Reviewer #5

- This manuscript presents a new methodology for assessing the fairness of income distribution in Thailand. It utilizes the concepts of distributive and procedural justice to quantify what is considered fair inequality, analyzing whether specific income brackets are receiving more or less than their fair share. This research could aid in evaluating the effectiveness of income redistribution policies and in formulating strategies for achieving fair income distribution.

Following the suggestions of Reviewers 1 and 2, the authors have made significant improvements to the paper. These enhancements include clarifying the study's importance and implications, as well as providing detailed explanations of the research methodology and data.

Additional Reviewer Comments:

However, the following further improvements are necessary:

Clear Definition of Distributive Justice Required: The paper mentions distributive justice but lacks a specific definition or theoretical basis. Since various definitions and interpretations of distributive justice exist, the authors must clarify which definition they are using and the rationale behind this choice. This clarification is important for understanding the research methodology and conclusions.

In response to Reviewer #5’s comment on the lack of definition of distributive justice, we provide the definition of distributive justice in Introduction, paragraph 9 in our revised manuscript.

- Improvement in Graphics Resolution: The resolution of graphs needs to be enhanced to more clearly convey the data and analysis results. This improvement is vital for effectively communicating the research findings.

We would like to inform Reviewer #5 that we prepared all figures by using a tool called PACE as recommended in PLOS ONE’s guideline for preparing figure. According to our understanding, the resolution of graphs is not clear because it is converted into PDF format when our manuscript was sent out for peer review. 

- The methodology and results proposed in this paper have the potential to enhance the understanding of fair income distribution and could provide valuable insights to academia and policy-makers. Considering the above aspects, this manuscript appears to have sufficient potential for publication in PLOS ONE.

We would like to sincerely thank Reviewer #5 for finding our manuscript useful and recommending it for publication in PLOS ONE.

---

## [Decision Letter · Decision Letter 2]

20 Mar 2024

Quantifying fair income distribution in Thailand

PONE-D-23-25866R2

Dear Dr. Sitthiyot,

We’re pleased to inform you that your manuscript has been judged scientifically suitable for publication and will be formally accepted for publication once it meets all outstanding technical requirements.

Kind regards,

Stephen Esaku

Academic Editor

PLOS ONE

Additional Editor Comments (optional):

Reviewers' comments:

Reviewer's Responses to Questions

**Comments to the Author**

1. If the authors have adequately addressed your comments raised in a previous round of review and you feel that this manuscript is now acceptable for publication, you may indicate that here to bypass the “Comments to the Author” section, enter your conflict of interest statement in the “Confidential to Editor” section, and submit your "Accept" recommendation.

Reviewer #3: (No Response)

Reviewer #5: All comments have been addressed

2. Is the manuscript technically sound, and do the data support the conclusions?

Reviewer #3: Yes

Reviewer #5: Yes

3. Has the statistical analysis been performed appropriately and rigorously? 

Reviewer #3: Yes

Reviewer #5: Yes

4. Have the authors made all data underlying the findings in their manuscript fully available?

Reviewer #3: Yes

Reviewer #5: Yes

5. Is the manuscript presented in an intelligible fashion and written in standard English?

Reviewer #3: Yes

Reviewer #5: Yes

6. Review Comments to the Author

Reviewer #3: In general, all corrections have been made. The authors tried to fix everything. I hope that this work will make a contribution.

Reviewer #5: The authors have well understood the concerns I raised in my previous review and have effectively revised the manuscript accordingly. These revisions have helped improve the quality of the manuscript. I believe the manuscript is now ready for publication in PLOS ONE.

7. PLOS authors have the option to publish the peer review history of their article (what does this mean?). If published, this will include your full peer review and any attached files.

Reviewer #3: **Yes: **Anel Kireyeva

Reviewer #5: No

---

## [Editor Report · Acceptance letter]

25 Mar 2024

PONE-D-23-25866R2 

PLOS ONE

Dear Dr. Sitthiyot, 

I'm pleased to inform you that your manuscript has been deemed suitable for publication in PLOS ONE. Congratulations! Your manuscript is now being handed over to our production team.

Kind regards, 

on behalf of

Dr. Stephen Esaku 

Academic Editor

PLOS ONE